# Potentiating hypoxic microenvironment for antibiotic activation by photodynamic therapy to combat bacterial biofilm infections

Weijun Xiu[1], Ling Wan[1], Kaili Yang[1], Xiao Li[1], Lihui Yuwen [1✉], Heng Dong [2], Yongbin Mou[2], Dongliang Yang [3] & Lianhui Wang [1✉]

Traditional antibiotic treatment has limited efficacy for the drug-tolerant bacteria present in biofilms because of their unique metabolic conditions in the biofilm infection microenvironment. Modulating the biofilm infection microenvironment may influence the metabolic state of the bacteria and provide alternative therapeutic routes. In this study, photodynamic therapy is used not only to eradicate methicillin-resistant *Staphylococcus aureus* biofilms in the normoxic condition, but also to potentiate the hypoxic microenvironment, which induces the anaerobic metabolism of methicillin-resistant *Staphylococcus aureus* and activates the antibacterial activity of metronidazole. Moreover, the photodynamic therapy-activated chemotherapy can polarize the macrophages to a M2-like phenotype and promote the repair of the biofilm infected wounds in mice. This biofilm infection microenvironment modulation strategy, whereby the hypoxic microenvironment is potentiated to synergize photodynamic therapy with chemotherapy, provides an alternative pathway for efficient treatment of biofilm-associated infections.

[1] State Key Laboratory for Organic Electronics and Information Displays & Jiangsu Key Laboratory for Biosensors, Institute of Advanced Materials (IAM), Jiangsu National Synergetic Innovation Centre for Advanced Materials (SICAM), Nanjing University of Posts and Telecommunications, Nanjing 210023, China. [2] Nanjing Stomatological Hospital, Medicine School of Nanjing University, Nanjing 210008, China. [3] School of Physical and Mathematical Sciences & Institute of Advanced Materials (IAM), Nanjing Tech University, Nanjing 211800, China. ✉email: iamlhyuwen@njupt.edu.cn; iamlhwang@njupt.edu.cn

Bacterial biofilm-related infections affect millions of people worldwide and pose a serious threat to the public healthcare system[1–5]. In biofilms, bacteria are packed together by self-secreted extracellular polymeric substances (EPS)[6]. Shielded by such compact frameworks, bacteria can develop antibiotic tolerance because of their varied metabolic states to adapt to the biofilm infection microenvironment (BIM)[1,7–12]. On the other hand, bacteria in biofilms can avoid the immune attack under the protection of EPS. The infiltrating immune cells around bacterial biofilms produce excessive inflammatory factors, which causes the infected tissues to remain in the inflammatory phase and impede the normal healing process[13–16]. Consequently, the bacterial biofilms are usually associated with recalcitrant infections and serious inflammation. Clinically, high-dose antibiotics and surgical resection are the primary choices to treat bacterial biofilm infections; however, they are frequently ineffective or too painful[17,18]. Therefore, an effective therapeutic strategy for bacterial biofilm infections remains an urgent need.

Owing to the complicated interactions between the bacteria and host, the biofilm-infected tissues usually have a unique microenvironment, such as low pH, hypoxia, specific enzymes, lack of nutrients, and so on[10,19]. In bacterial biofilms, the $O_2$ level gradually decreases from the outer layer to the inner layer owing to the limitation of $O_2$ diffusion in biofilm. Importantly, the BIM greatly influences the metabolic state of bacteria, and is closely related to the therapeutic outcome of antibiotics[20,21]. To adapt to the complex BIM, bacteria in biofilms can develop diverse phenotypes with various metabolic states and drug susceptibility[10,20–22]. In the outer layer of biofilm, the bacteria exhibit metabolically active state owing to sufficient oxygen, while the bacteria located in the inner layer of biofilm with limited oxygen supply usually display metabolically less active state. Although antibiotics can kill most metabolically active bacteria in biofilms, metabolically less active bacteria that are deeply located in biofilms have high antibiotic tolerance, which causes incomplete bacteria-killing and the relapse of biofilm infections[7,10,12,23]. Hence, the BIM has a close relationship with the efficiency of antibiotics, and the modulation of the BIM may provide opportunities for managing bacterial biofilm infections.

Herein, we develop the hypoxia-potentiating strategy by combining the photodynamic therapy (PDT) and the prodrug metronidazole (MNZ) to treat bacterial biofilm infections. As shown in Fig. 1a, the hyaluronic acid (HA) was functionalized with chlorin e6 (Ce6) and MNZ to form HA-Ce6-MNZ nanoparticles (HCM NPs). Following delivery into methicillin-resistant *Staphylococcus aureus* (MRSA) biofilm infected sites, HCM NPs are decomposed to release Ce6 and MNZ by hyaluronidase (Hyal) secreted from MRSA. During laser irradiation, Ce6 can generate $^1O_2$ and kill the bacteria in biofilms under normoxic conditions. The depletion of $O_2$ by PDT subsequently potentiates hypoxia in biofilms and promotes the generation of nitroreductase by MRSA, which can further reductively activate MNZ and kill the metabolically less active bacteria under the hypoxic condition. Such PDT-activated chemotherapy can be used to eradicate MRSA biofilms, and macrophages in infected tissues can be further polarized to a M2-like phenotype to facilitate tissue healing (Fig. 1b). The therapeutic efficacy of HCM NPs was studied in subcutaneous MRSA biofilm-infected mice. Moreover, in situ sprayed fibrin gel containing HCM NPs was demonstrated to be effective in treating MRSA biofilm-infected chronic wounds in diabetic mice by PDT-activated chemotherapy. This work highlights a BIM-modulation strategy to efficiently combat bacterial biofilm-related infections.

## Results
### Preparation and characterization of HCM NPs.
In this study, HA-Ce6 was synthesized by conjugating Ce6 to amine-functionalized HA (A-HA), which can self-assemble into HA-Ce6 NPs (HC NPs)[24]. Then, MNZ was loaded into HC NPs to form HCM NPs (Fig. 2a). As shown in Fig. 2b–d, HCM NPs have a spheroid morphology similar to HC NPs, and increased hydrodynamic diameter. The polydispersity index of HC NPs and HCM NPs is approximately 0.22 and 0.31, respectively. The zeta potential results show that both HC NPs and HCM NPs have negative charges (Supplementary Fig. 1), indicating the successful functionalization of A-HA. As shown in Fig. 2e, the infrared (IR) absorption bands at 1485 cm$^{-1}$ and 1716 cm$^{-1}$ can be assigned to the stretching vibration of N=O (−NO$_2$) from MNZ and the bending vibration of C=O from Ce6, respectively[25,26]. Besides, the characteristic peaks near 400 nm and 660 nm for Ce6, and 320 nm for MNZ emerged in the absorption spectrum of HCM NPs, demonstrating the successful preparation of HCM NPs (Fig. 2f).

As MRSA can secrete Hyal and decompose HA[3], Hyal in MRSA biofilms can be used to trigger the drug release of HCM NPs (Fig. 3a). First, the Hyal-responsive release of Ce6 and MNZ was studied. In HCM NPs, the fluorescence of Ce6 was quenched because of its aggregated form (Supplementary Fig. 2)[27]. Figure 3b and c indicate that the fluorescence of HCM NPs recovered with the presence of Hyal under acidic conditions (pH 5.5). The release efficiency of Ce6 and MNZ from HCM NPs was approximately 80% and 78%, respectively, after 24 h incubation with Hyal (Fig. 3d and e). As illustrated in Fig. 3f and g, the fluorescence signal of Ce6 showed significant recovery after the incubation of HCM NPs with MRSA biofilms, while that of HCM NPs incubated with mammalian cells (mice smooth muscle cells, SMCs) showed a negligible increase. After incubating for 24 h, the fluorescence signal of Ce6 in MRSA biofilms was approximately 2-fold higher than that incubated with SMCs (Supplementary Fig. 3), indicating that HCM NPs can effectively release the loaded drugs in MRSA biofilms. Besides, MRSA biofilms showed limited change of average thickness after incubated with HCM NPs for 24 h (Supplementary Fig. 4). Hyal-induced decomposition of HCM NPs was also confirmed by DLS and TEM (Fig. 3h and i). Besides, the $^1O_2$ generation property of HCM NPs under light irradiation was measured by using 9, 10-anthracenediyl-bis(methylene)-dimalonic acid (ABDA) as an indicator[28]. As shown in Fig. 3j, the $^1O_2$ generation of HCM NPs in the presence of Hyal was higher than that of other groups, indicating their Hyal-responsive photodynamic effect.

Before further biomedical application studies, the colloidal stability and cytotoxicity of HCM NPs were evaluated. HCM NPs showed no obvious aggregation and change of diameter after dispersed in different mediums (H$_2$O, phosphate-buffered saline (PBS), and minimum Eagle's medium (MEM)) for 24 h, indicating good colloidal stability (Supplementary Fig. 5). As shown in Supplementary Fig. 6, the viability of human normal liver (L-O2) cells exceeded 90% after being incubated with HCM NPs (Ce6: 160 µg/mL) for 1 d. In contrast, the viability of L-O2 cells decreased below 65% after incubation with HCM NPs (Ce6: 160 µg/mL) in the medium containing Hyal.

### Enhanced in vitro anti-biofilm effect of HCM NPs by potentiating hypoxia.
As a facultative anaerobe, *S. aureus* can grow using either aerobic respiration in normoxic conditions or using nitrate and mixed fermentation under the hypoxic condition[29,30]. In biofilms, bacteria exhibit various metabolic states owing to the complex BIM, which has significant implications for the therapeutic efficiency of antibiotics. Although the facultative bacteria in biofilms with the normoxic condition can be effectively inactivated by antibiotics, the bacteria in biofilms under the hypoxic condition usually show significant drug tolerance due to their anaerobic metabolic state[31]. As a prodrug, MNZ could be

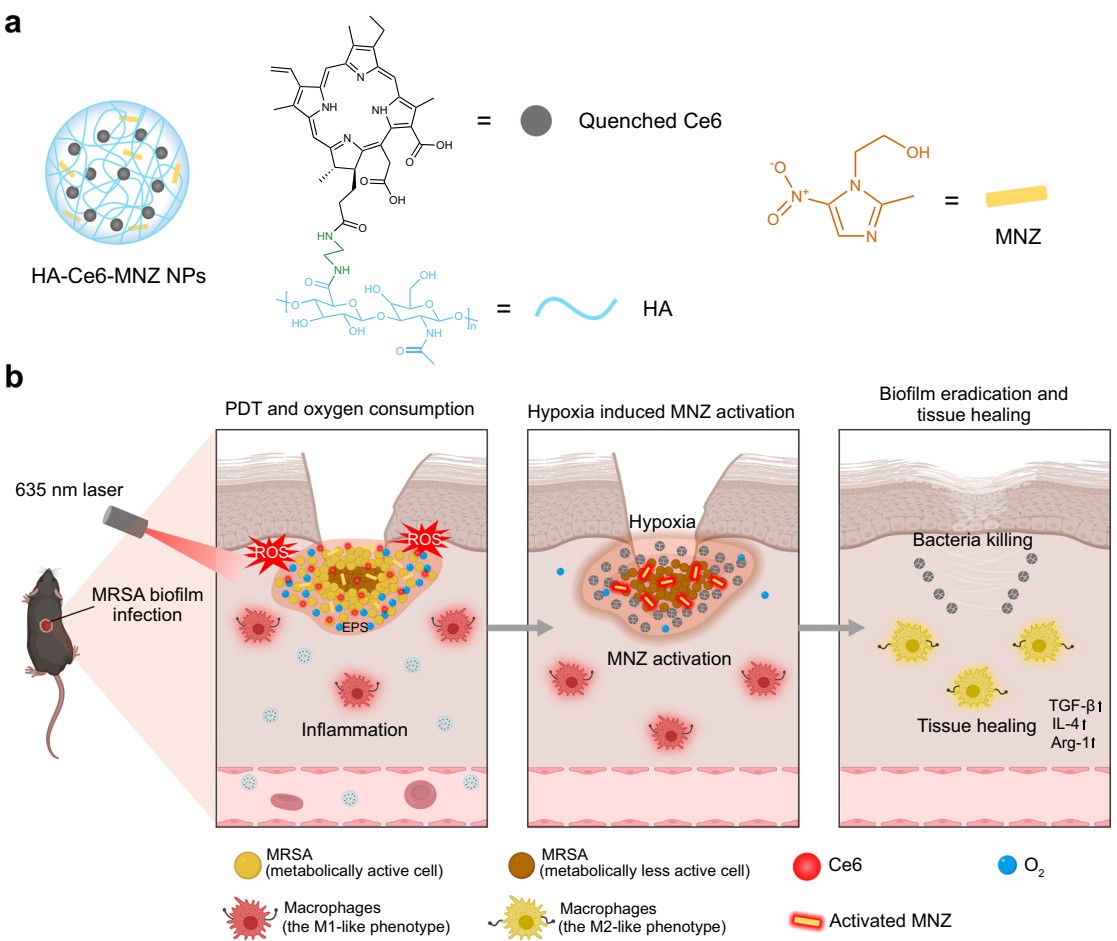

**Fig. 1 Potentiating hypoxia by PDT for antibiotic activation to combat MRSA biofilm infections. a** Schematic illustration of the structure of HA-Ce6-MNZ nanoparticles (HCM NPs, hyaluronic acid (HA), chlorin e6 (Ce6), metronidazole (MNZ)). **b** The therapeutic processes of MRSA biofilms, including the release of Ce6 and MNZ, PDT-potentiated hypoxia, activation of MNZ, elimination of bacterial biofilms, and infected tissue healing.

activated with the assistance of electron transfer proteins, such as nitroreductase, in the cytoplasm of bacteria under low redox potential[32–34]. Amine derivatives reduced from MNZ can induce the damage of DNA helix and cause bacterial death. Although MNZ is commonly used to treat anaerobe infections, it has also been reported to have limited effects on facultative anaerobes[35–37]. In the hypoxic BIM, MRSA can overexpress nitroreductase (Supplementary Fig. 7), which may activate the antibacterial activity of MNZ for killing facultative anaerobes[38,39]. As shown in Supplementary Fig. 8 and Supplementary Fig. 9, the viability of MRSA showed a negligible decrease following treatment with MNZ under normoxic conditions, while approximately 25% of MRSA were killed by MNZ under hypoxic conditions. The combination of MNZ with oxygen-consuming PDT could work synergistically; therefore, we hypothesized that PDT could inactivate the bacteria under normoxic conditions, deplete $O_2$ in the local environment, potentiate the hypoxic condition within biofilms, promote the expression of nitroreductase by bacteria, and finally activate MNZ to kill the remaining bacteria inside biofilms (Fig. 4a).

To prove the above hypothesis, the anti-biofilm effect of HCM NPs was evaluated by MRSA biofilms in vitro. As shown in Fig. 4b, MRSA biofilms showed a significantly decreased level of $O_2$ following the treatment of HCM NPs with laser irradiation, indicating that PDT could potentiate the hypoxia level of biofilms. As indicated in Fig. 4c and d, the MRSA inactivation efficiency for HC NPs with laser irradiation (3.1 log, 99.92%) was much higher than that of vancomycin (Van, 0.7 log, 78.3%) or

MNZ (0.2 log, 12.3%). In contrast, HCM NPs with laser irradiation can inactivate MRSA by 5.9 log (99.9998%), much higher than PDT or antibiotic treatment alone, which demonstrates the effectiveness of PDT-activated chemotherapy. Besides, the MRSA biofilm showed significant structural damage and decreased biofilm biomass after treatment of HCM NPs with laser irradiation (Fig. 4e and f). The SEM images further indicate the excellent therapeutic efficacy of HCM NPs (Fig. 4g).

To investigate the influence of PDT on the MNZ penetration, the permeability of MNZ in MRSA biofilms with or without PDT was studied by using Transwell inserts (Supplementary Fig. 10a). As shown in Supplementary Fig. 10b, about 74.3% of MNZ was collected in receiver plate after HCM NPs incubated with MRSA biofilms without laser irradiation, while about 74.5% of MNZ was collected for the group with laser irradiation, suggesting the PDT process had limited influence on the penetration of MNZ within MRSA biofilms. Besides, MNZ displayed much higher bacterial inactivation efficiency under hypoxic conditions than the normoxic condition (Supplementary Fig. 11), indicating the potentiated hypoxic microenvironment can significantly improve the antibacterial activity of MNZ in biofilms. The above results indicate that PDT-potentiated hypoxia in MRSA biofilms can change the metabolism of MRSA and further activate MNZ for better inactivation of MRSA in biofilms. In addition, the penetration of MNZ was not significantly enhanced by the PDT process, which showed limited contribution to the improved inactivation efficiency of MRSA by MNZ in biofilms.

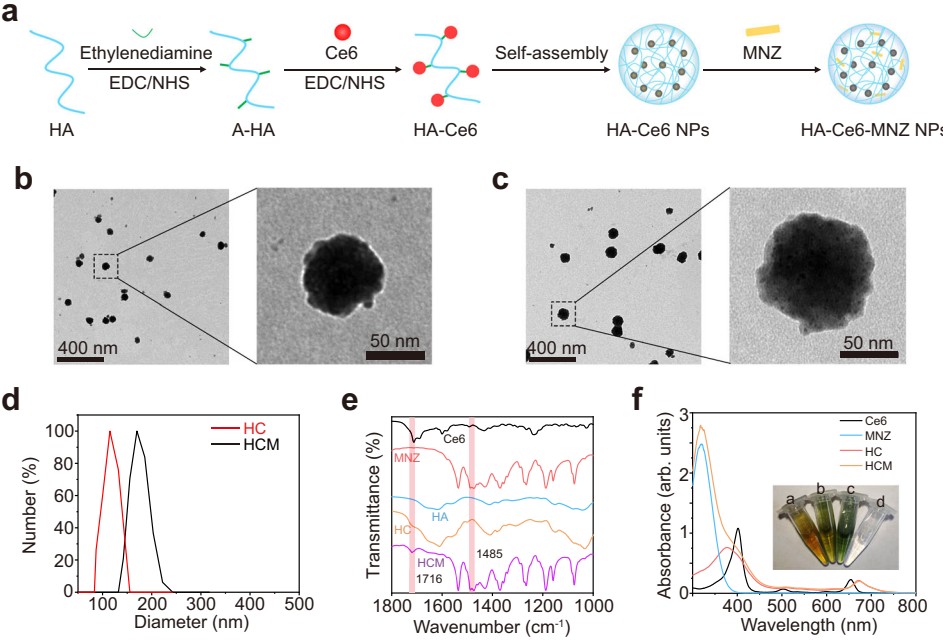

**Fig. 2 Preparation and characterization of HCM NPs. a** Schematic illustration of the preparation of HCM NPs (amine-functionalized HA (A-HA), 1-(3-dimethylaminopropyl)-3-ethylcarbodiimide hydrochloride (EDC), N-hydroxysulfosuccinimide sodium salt (NHS)). Transmission electron microscopy (TEM) images of HA-Ce6 nanoparticles (HC NPs) (**b**) and HCM NPs (**c**) with low magnification (left) and high magnification (right). Three independent experiments were performed and representative results are shown in **b** and **c**. **d** Hydrodynamic sizes of HC NPs and HCM NPs measured by dynamic light scattering (DLS). **e** Fourier transformed infrared (FT-IR) spectra of Ce6, MNZ, HA, HC NPs, and HCM NPs. **f** Ultraviolet-visible-near infrared (UV-vis-NIR) spectra of Ce6, MNZ, HA, HC NPs, and HCM NPs. Inset: photographs of different solutions (a: HC NPs; b: HCM NPs; c: Ce6; d: MNZ). Source data are provided as a Source Data file.

The long-term anti-biofilm effect was studied to evaluate the therapeutic effectiveness of PDT-activated chemotherapy. From Supplementary Fig. 12a and b, the live bacteria in MRSA biofilms treated by HCM NPs with laser irradiation (HCM + L) showed no significant increase at 72 h post-treatment under eutrophic conditions, while the bacteria in other groups showed obvious regrowth. Moreover, the micrographs of MRSA biofilms stained by crystal violet in the HCM + L group showed that the biofilm structure was completely destructed without regrowth in 3 d after treatment (Supplementary Fig. 12c), which was not achieved by other treatments. Therefore, PDT-activated chemotherapy could completely eradicate MRSA biofilms and prevent their regrowth.

**Treatment of subcutaneous MRSA biofilm-infected mice**. To evaluate the in vivo anti-biofilm efficacy of HCM NPs, the subcutaneous MRSA biofilm-infected mice model was constructed (Fig. 5a)[40]. The enhanced expression of biofilm-associated genes verified the formation of MRSA biofilms in infected tissues (Supplementary Fig. 13)[41]. The wound blotting result also indicated the formation of MRSA biofilms in this model (Supplementary Fig. 14)[42]. As shown in Fig. 5b, the fluorescence signal appeared in MRSA biofilm infected sites after in situ injection of HCM NPs, while a limited fluorescence signal was observed for normal tissues. HCM NPs were also intravenously (i.v.) injected into MRSA biofilm-infected mice. The fluorescence signal from Ce6 was obviously increased in MRSA biofilm-infected tissues and peaked at 12 h post-injection (Fig. 5c and Supplementary Fig. 15). Meanwhile, the fluorescence images of the biofilm-infected tissue and major organs harvested from mice also demonstrate similar results, suggesting efficient accumulation and drug release of HCM NPs in biofilm-infected tissues (Supplementary Fig. 16)[19,43].

The hypoxia level of infected tissues after various treatments was further evaluated by using immunofluorescence staining of hypoxia inducible factor-1α (HIF-1α). As shown in Fig. 5d–f, the overexpression of HIF-1α and vascular endothelial growth factor (VEGF) in HC + L and HCM + L groups indicates that PDT could potentiate the hypoxia level in biofilm infected tissues. As shown in Fig. 5g and h, the infected tissue almost disappeared at 8 d post-treatment in the HCM + L group, while that in other groups showed a limited reduction. Moreover, the bacterial inactivation efficiency was approximately 6.4 log (99.99996%) in the HCM + L group, which was much better than that in other groups (Fig. 5i and j). These results indicate the excellent in vivo anti-biofilm efficacy of PDT-activated chemotherapy. As shown in Fig. 5k, the infected tissues of mice in HCM + L group exhibit little inflammatory cell infiltration and high extent of collagen deposition, while other groups still show significant inflammation. Besides, the body weight and major organs of the mice in HCM + L group showed no observable abnormities, indicating low toxicity of HCM NPs (Supplementary Fig. 17).

**Macrophage polarization after PDT-activated chemotherapy**. For biofilm-related infections, the long-term existence of bacteria causes a persistent inflammatory state in the infected tissues, and seriously disturbs the normal repair process[13]. Macrophages perform critical regulatory functions in different tissue repair stages by polarization[44–46]. M1-like macrophages are associated with inflammation in the infected tissues, while M2-like macrophages have key roles in tissue repair[44,47]. Hence, the influence of PDT-activated chemotherapy on macrophage polarization in tissues was further evaluated. As shown in Fig. 6a–c, the infected tissues of mice in HCM + L groups showed decreased M1-like macrophages (F4/80$^+$ and CD80$^+$) infiltration and increased M2-like macrophages (F4/80$^+$ and CD206$^+$) infiltration compared

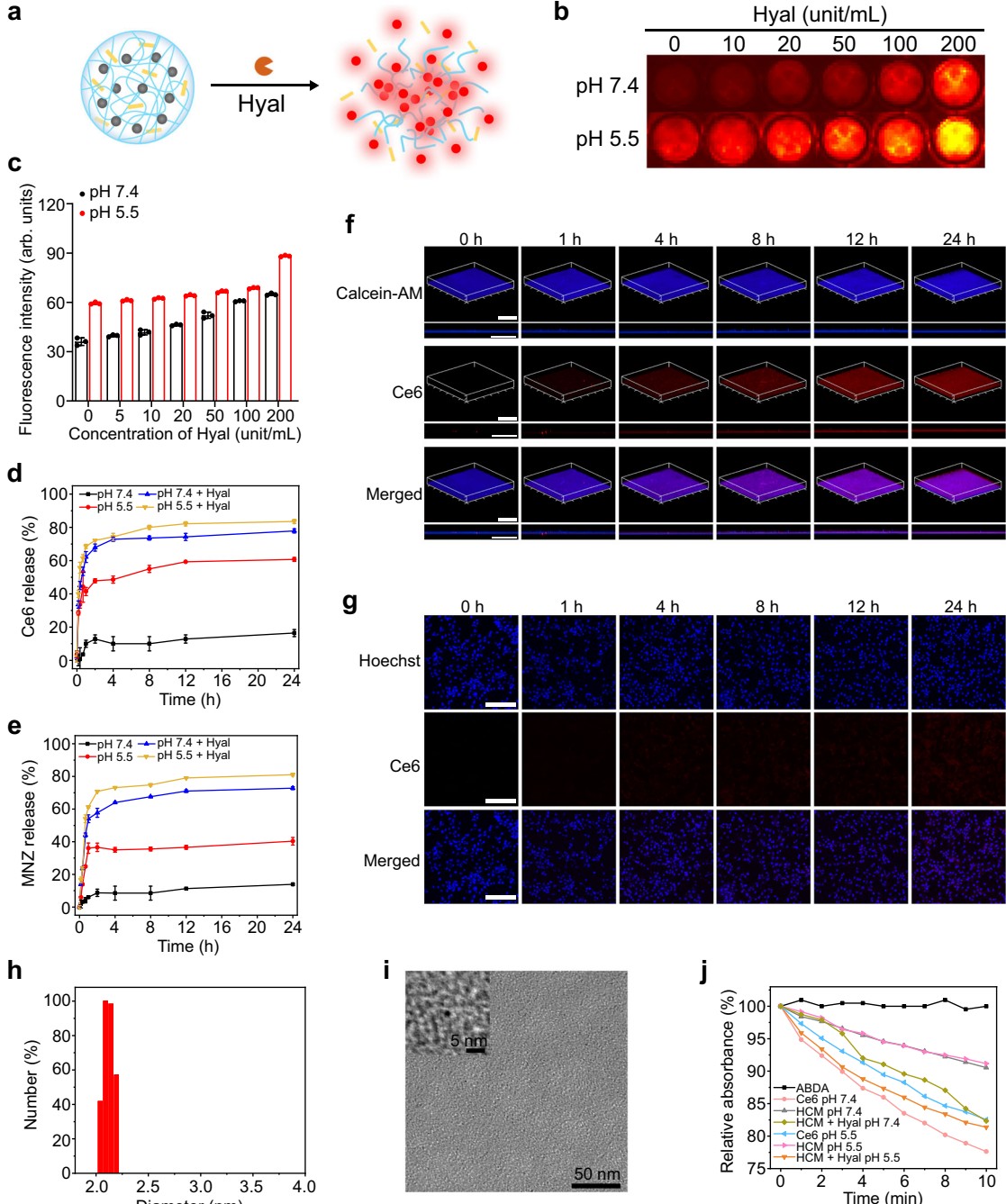

**Fig. 3 Hyal-responsive drug release of HCM NPs. a** Scheme of the decomposition of HCM NPs in presence of Hyal. Fluorescence images (**b**) and corresponding fluorescence intensity (**c**) of HCM NPs (Ce6: 40 μg/mL) after incubation with Hyal at different concentrations for 12 h ($n = 3$ independent samples; mean ± SD). Release of Ce6 (**d**) and MNZ (**e**) from HCM NPs (Ce6: 40 μg/mL; MNZ: 20 μg/mL) under different conditions ($n = 3$ independent samples; mean ± SD). The confocal laser scanning microscopy (CLSM) images of MRSA biofilms (**f**) and SMCs (**g**) after incubation with HCM NPs (Ce6: 50 μg/mL; MNZ: 25 μg/mL) for different times. Scale bar is 200 μm. Hydrodynamic size (**h**) and TEM images (**i**) of HCM NPs after incubation with Hyal (100 unit/mL) for 24 h. **j** Relative absorbance of ABDA at 380 nm ($OD_{380}$) after incubation at various conditions under laser irradiation (635 nm, 20 mW/cm$^2$) for different times. Three independent experiments were performed and representative results are shown in **f**, **g**, and **i**. Source data are provided as a Source Data file.

with the other groups. Furthermore, the lower level of interleukin-12p70 (IL-12p70) and a higher level of interleukin-4 (IL-4) in HCM + L group than in other groups suggest that the macrophages were polarized to M2 phenotype after PDT-activated chemotherapy (Fig. 6d and e). This macrophage polarization was further confirmed by the high expression of

CD206 and the low expression of CD80 in infected tissues in HCM + L group (Fig. 6f and g). The increase of tumor necrosis factor-α (TNF-α), IL-12p70, and IL-6, and the decrease of transforming growth factor-β (TGF-β), IL-4, and arginase-1 (Arg-1) also suggest the M2 polarization in HCM + L group (Fig. 6h and i). Therefore, PDT-activated chemotherapy could be used to

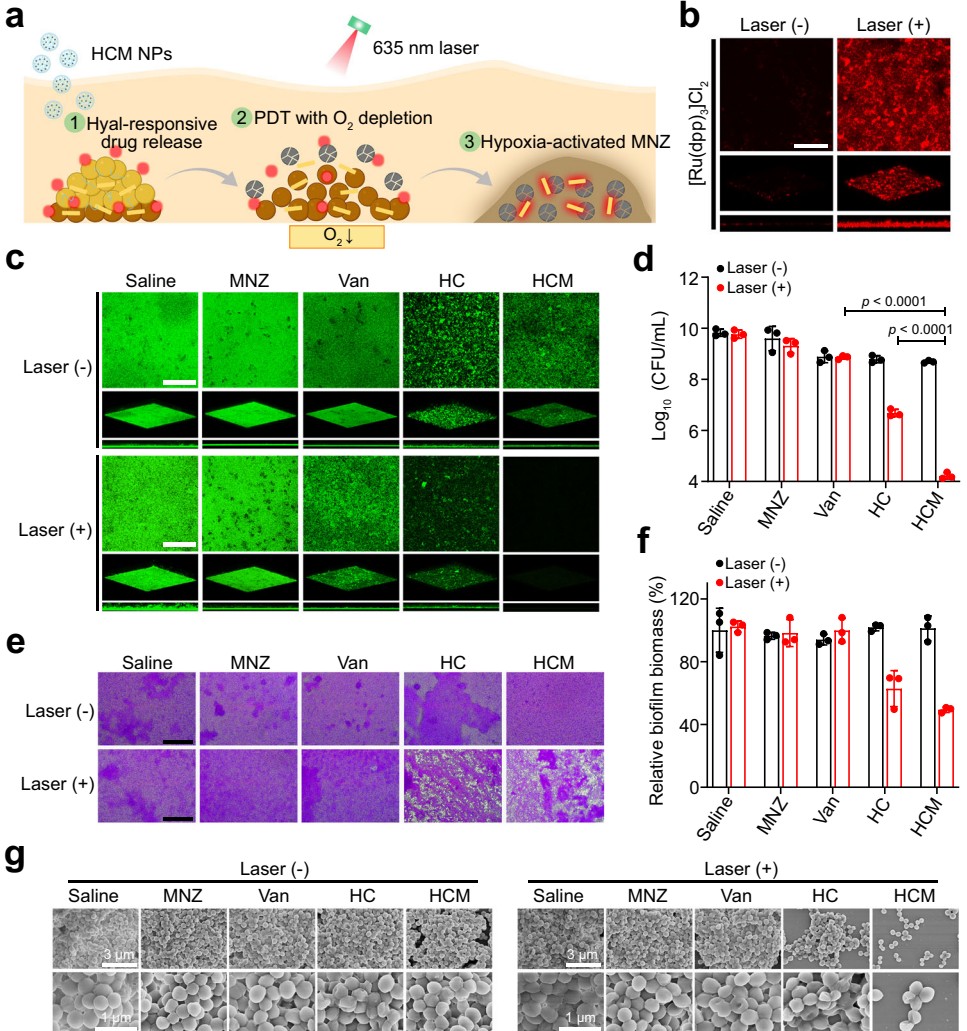

**Fig. 4 PDT-activated chemotherapy of MRSA biofilms by hypoxia potentiation. a** Schematic illustration of the PDT-induced hypoxia and subsequent activation of MNZ for enhanced anti-biofilm treatment by HCM NPs. **b** Three-dimensional (3D) CLSM images of $[Ru(dpp)_3]Cl_2$ stained MRSA biofilms with or without PDT (635 nm, 20 mW/cm², 30 min) by HCM NPs. Scale bar is 200 μm. Size of CLSM images is 630 μm × 630 μm. **c** 3D CLSM images of MRSA biofilms stained by Calcein-AM (green) after various treatments. Size of CLSM images is 630 μm × 630 μm. Scale bar is 200 μm. **d** Numbers of viable MRSA within biofilms after various treatments ($n = 3$ biologically independent samples; mean ± SD). Statistical significance was analyzed via two-way ANOVA test with a Tukey post-hoc test. **e** Micrographs of MRSA biofilms stained by crystal violet after various treatments. Scale bar is 200 μm. **f** Relative biofilm biomass of MRSA biofilms after various treatments ($n = 3$ biologically independent samples; mean ± SD). **g** Scanning electron microscopy (SEM) images of MRSA biofilms after various treatments. Three independent experiments were performed and representative results are shown in **b**, **c**, **e**, and **g**. Source data are provided as a Source Data file.

polarize macrophages to an anti-inflammatory phenotype and promote tissue healing.

**Treatment of MRSA biofilm-infected wounds in diabetic mice.** Nowadays, up to 25% of patients with diabetics confront a lifetime risk of chronic wounds, which may lead to amputation and even death[48]. One of the major reasons for the therapeutic failure of diabetic chronic wounds is the formation of bacterial biofilms, which often develop superimposed infections and delay the repair process[49]. Here, we prepared HCM NPs-based fibrin gel (HCM gel) to treat MRSA biofilm-infected wounds in diabetic mice. As shown in Fig. 7a and b, the HCM gel was formed in situ after sequentially spraying HCM NPs-containing fibrinogen solution and thrombin solution on the wound bed. The HCM NPs showed limited changes of the hydrodynamic size and zeta potential after forming HCM gel (Supplementary Fig. 18). CLSM images of the HCM gel demonstrate that the distribution of the fluorescence

signal from Ce6 is consistent with that from fibrin, suggesting the successful encapsulation of HCM NPs in the fibrin gel (Fig. 7c).

To evaluate the in vitro drug release capability, HCM gel was incubated with or without Hyal (200 unit/mL) under different pH conditions. As illustrated in Fig. 7d and e, Ce6 and MNZ were gradually released under acidic conditions in presence of Hyal. After in situ formation of the HCM gel, the fluorescence signal gradually increased in MRSA biofilm infected wounds, while the fluorescence signal of the normal wounds was relatively unchanged (Fig. 7f and g), indicating that the HCM gel can release the loaded drugs in the BIM. Then, the anti-biofilm efficacy of the HCM gel by PDT-activated chemotherapy was evaluated in diabetic mice (Fig. 8a). Following 12 d of treatment, the infected wounds in HCM + L group completely disappeared, while those in other groups still remained unhealed (Fig. 8b and c). Notably, the HCM + L group displayed the best bacterial inactivation efficiency (8.0 log, 99.999999%), which was much higher than that shown by HC group (4.3 log, 99.994%), HCM

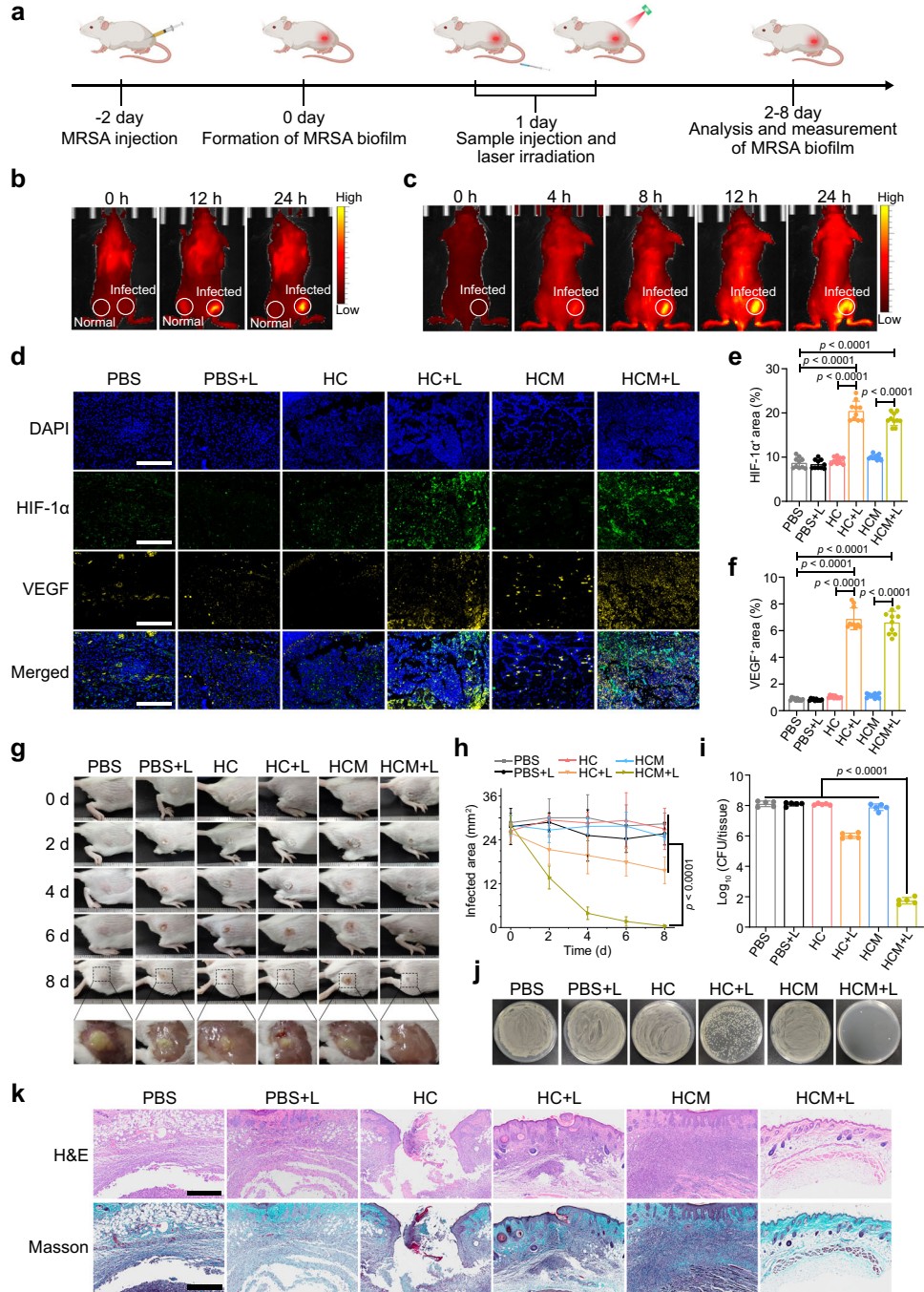

**Fig. 5 Treatment of subcutaneous MRSA biofilm infected mice by HCM NPs. a** Schematic illustration of the experimental procedure for treating MRSA biofilm infected mice. **b** Fluorescence images of MRSA biofilm infected mice after in situ injection of HCM NPs (Ce6: 100 μg/mL, 50 μL) in normal tissues (left side of thigh) and biofilm infected tissues (right side of thigh), respectively. **c** Fluorescence images of MRSA biofilm infected mice after *i.v.* injection of HCM NPs (Ce6 = 4 mg/kg; MNZ = 2 mg/kg). **d** Representative immunofluorescence images of HIF-1α (green) and VEGF (yellow) in MRSA biofilm infected tissues after various treatments for 4 d. Scale bar is 400 μm. The percentage of HIF-1α$^+$ (**e**) and VEGF$^+$ (**f**) area in various treatment groups calculated from the immunofluorescence images ($n = 10$ biologically independent samples; mean ± SD). Photographs of the infected tissues (**g**) and infected area (**h**) of the mice after various treatments ($n = 5$ biologically independent samples; mean ± SD). **i** Quantification of viable bacteria inside biofilm-infected tissues at 8 d post-treatment ($n = 5$ biologically independent samples; mean ± SD). **j** Photographs of MRSA colonies from infected tissues at 8 d post-treatment. **k** H&E and Masson's trichrome stained slices of the infected tissues from mice at 8th d post-treatment. Scale bar is 500 μm. Statistical significance was analyzed via one-way ANOVA with a Tukey post-hoc test. Source data are provided as a Source Data file.

group (4.9 log, 99.999%), and HC + L group (6.7 log, 99.99997%) (Fig. 8d and Supplementary Fig. 19).

The overexpression of HIF-1α in HCM + L group indicates that PDT can exacerbate hypoxia level of the biofilm-infected tissues (Supplementary Fig. 20). The immunofluorescence images

reveal that the M1-like macrophages decreased and the M2-like macrophages increased in HCM + L group (Fig. 8e–g), suggesting macrophage polarization to the M2-like phenotype in infected wounds. The decreased secretion of TNF-α and IL-12p70, and the increased secretion of Arg-1 and IL-4 in HCM + L group further

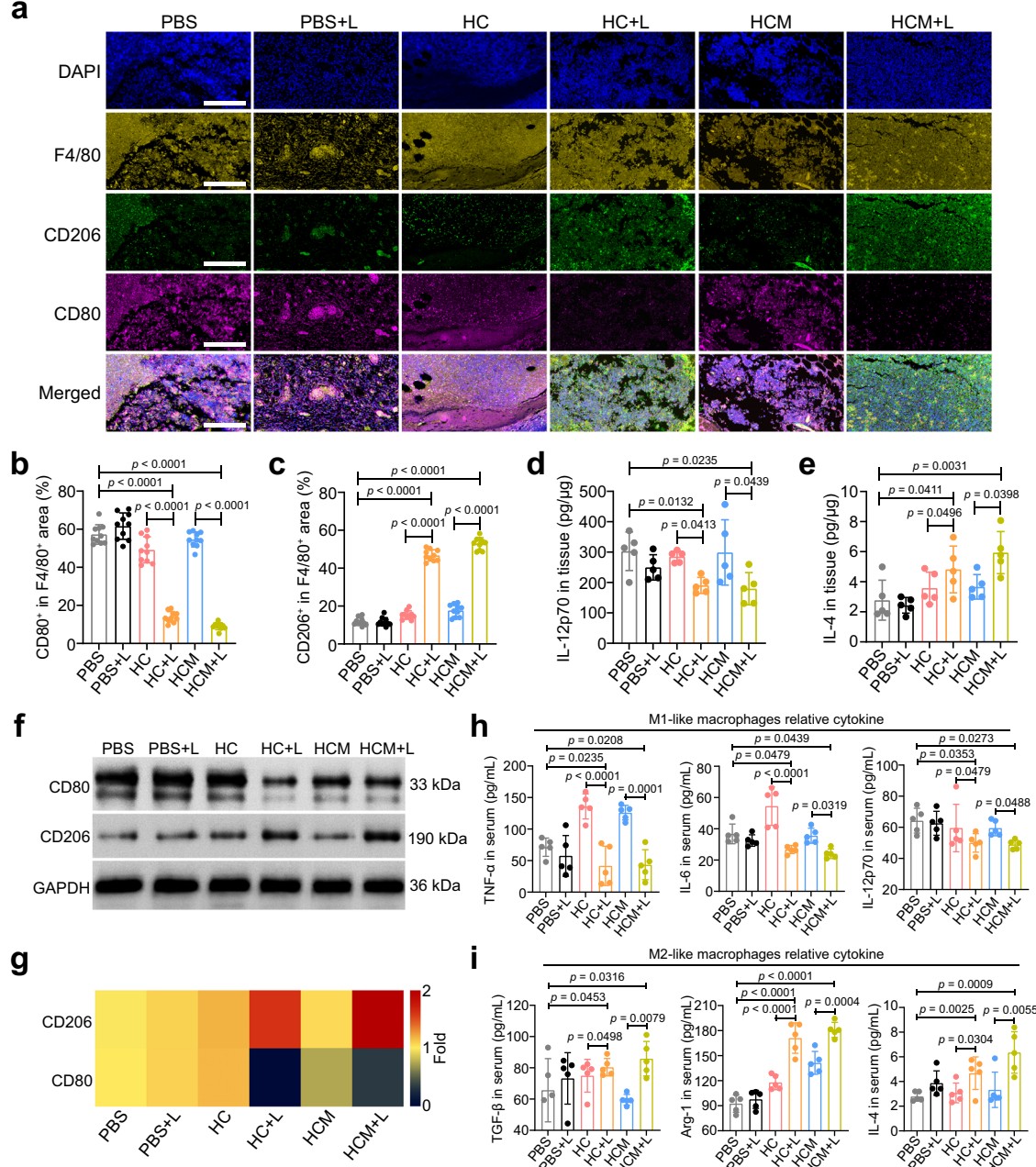

**Fig. 6 Macrophage polarization of MRSA biofilm infected mice after PDT-activated chemotherapy. a** Immunofluorescence images of F4/80 (yellow), CD206 (green), and CD80 (violet) in the infected tissues after treatment. Scale bar is 400 μm. Percentage of CD80+ (**b**) and CD206+ (**c**) area to total F4/80+ area calculated from the immunofluorescence images ($n = 10$ biologically independent samples; mean ± SD). Secretion levels of IL-12p70 (**d**) and IL-4 (**e**) in the infected tissues after treatment ($n = 5$ biologically independent samples; mean ± SD). **f** Expression levels of CD80 and CD206 in the infected tissues determined by western blotting after treatment. **g** Heat map of the relative expression of CD80 and CD206 in the infected tissues compared to the PBS group calculated from western blotting results. Secretion levels of M1 macrophage-relative cytokines (**h**) and M2 macrophage-relative cytokines (**i**) in the serums isolated from mice after treatment ($n = 5$ biologically independent samples; mean ± SD). All statistical significance was analyzed via one-way ANOVA with a Tukey post-hoc test. Source data are provided as a Source Data file.

confirm the M2-like macrophage polarization (Supplementary Fig. 21). H&E and Masson's trichrome staining photographs show significant re-epithelialization, collagen deposition, and less inflammatory cell infiltration in the biofilm-infected tissues from HCM + L group, demonstrating better therapeutic efficacy of the PDT-activated chemotherapy (Fig. 8h). After being treated by the HCM gel with laser irradiation, the mice showed no decrease in body weight and no obvious inflammation in major organs (Supplementary Fig. 22). In addition, the infected wound of mice

showed no observable retention of HCM NPs at 6 d post-treatment (Supplementary Fig. 23).

## Discussion

Traditional antibiotic treatment is generally insufficient to completely eliminate the existing bacterial biofilms[2,11,12]. In contrast to bacteria in planktonic form, bacteria in biofilms possess heterogeneous physiological activity to adapt to the complex BIM, which greatly reduces the therapeutic effect of antibiotics. For

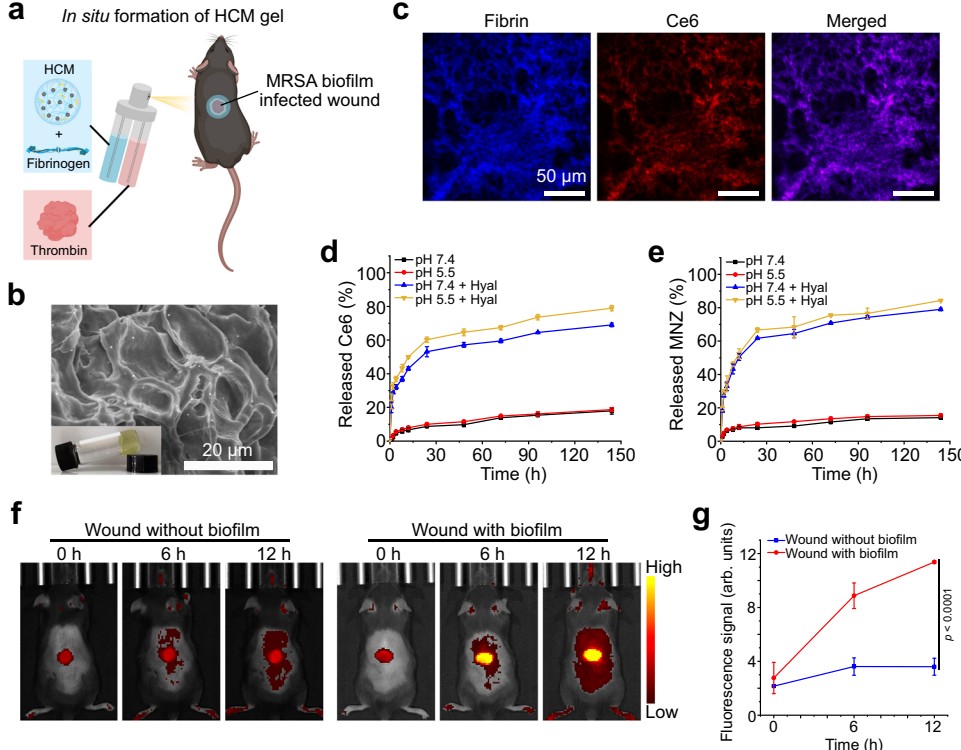

**Fig. 7 Preparation and characterization of the HCM gel. a** Schematic illustration of the HCM gel in situ formed on bacterial biofilm infected wounds. **b** SEM image of the HCM gel. Inset: photograph of HCM gel. **c** CLSM images of the HCM gel, in which the fibrinogen was labeled with fluorescein isothiocyanate (FITC, blue). Cumulative release profile of Ce6 (**d**) and MNZ (**e**) from the HCM gel incubated with or without Hyal under different pH conditions ($n = 3$ independent samples; mean ± SD). Fluorescence images (**f**) and fluorescence intensity (**g**) of normal wounds and wounds with MRSA biofilm infection in diabetic mice after treatment with HCM gel at different time points ($n = 3$ biologically independent animals from **f**; mean ± SD). Statistical significance was analyzed via one-way ANOVA with a Tukey post-hoc test. Three independent experiments were performed and representative results are shown in **b** and **c**. Source data are provided as a Source Data file.

instance, the bacteria in the outer layer of biofilms have an active metabolic state with high susceptibility to antibiotics, while the bacteria that locate deep within biofilms exhibit reduced metabolic activity and low susceptibility to antibiotics due to the limited oxygen and nutrients[2,10,19,50]. Besides, the bacteria in biofilms can escape the host immune attack due to the protection of EPS, which continuously induces inflammatory response and seriously impedes the tissue repair process[12,13,23,49]. Therefore, it is essential to completely eliminate bacteria in biofilms and mitigate the inflammation for the effective treatment of biofilm infections.

PDT is a promising therapeutic strategy for the treatment of bacterial infections. During the PDT process, photosensitizers can transfer $O_2$ to highly toxic reactive oxygen species (ROS), which can react with important biomolecules in cells and damage their functions. Compared with antibiotic treatment, PDT is much difficult for bacteria to develop resistance, which provides alternative solutions for drug-resistant bacteria infections[3,40,51]. Although PDT has several advantages compared with antibiotic treatment, the hypoxia in bacterial biofilm infection microenvironment significantly limits its therapeutic efficiency. To improve the antibacterial efficiency of PDT, previous works mainly focused on the strategy to relieve the hypoxia during PDT, while the strategy to potentiate the hypoxia PDT has not been exploited[52,53]. In our work, the hypoxic microenvironment in bacterial biofilms was potentiated by PDT to induce the anaerobic metabolism of MRSA and activate MNZ for bacteria-killing, which provides PDT-activated chemotherapy for the treatment of MRSA biofilm infections.

MNZ is an antimicrobial agent that has been widely used to treat anaerobic bacterial infections in clinic[37]. As a prodrug, MNZ could be reductively activated under low oxygen levels by forming imidazole fragments with high cytotoxicity toward anaerobic bacteria. Nevertheless, MNZ has been reported to have negligible efficacy for use in infections with facultative anaerobes[35,37]. As facultative anaerobes in hypoxic conditions can secrete electron transfer proteins, such as nitroreductase, which may transform MNZ to cytotoxic fragmentation[35–37], we hypothesized that MNZ could be activated under hypoxia caused by PDT to kill facultative anaerobes. As a second-generation photosensitizer, Ce6 can transfer $O_2$ into $^1O_2$ under laser irradiation, which can efficiently damage the vital biomolecules of tumor cells and bacteria. Ce6 has been approved by the U.S. Food and Drug Administration (FDA) for the PDT of esophageal, bladder, skin, head, and neck cancers[54]. Besides, Ce6 has also been studied as a PDT agent to treat bacterial infections, which can kill wide-spectrum of bacteria without the drug-resistant issues[51,55]. In this study, HCM NPs were developed as therapeutic nanoagents to treat MRSA biofilm infections by hypoxia-potentiation. During laser irradiation, the Ce6 generates $^1O_2$ and kills the MRSA in biofilms via the consumption of $O_2$, which subsequently potentiate the hypoxic condition within biofilms. MRSA in hypoxic BIM alter their metabolic state and generate electron transfer proteins for MNZ activation. The activated MNZ can further kill the bacteria in metabolically less active state within biofilms, which can not be achieved by antibiotic treatment or PDT alone. In biofilm-infected mice, bacteria inside biofilm-infected tissues were efficiently eliminated by PDT-activated

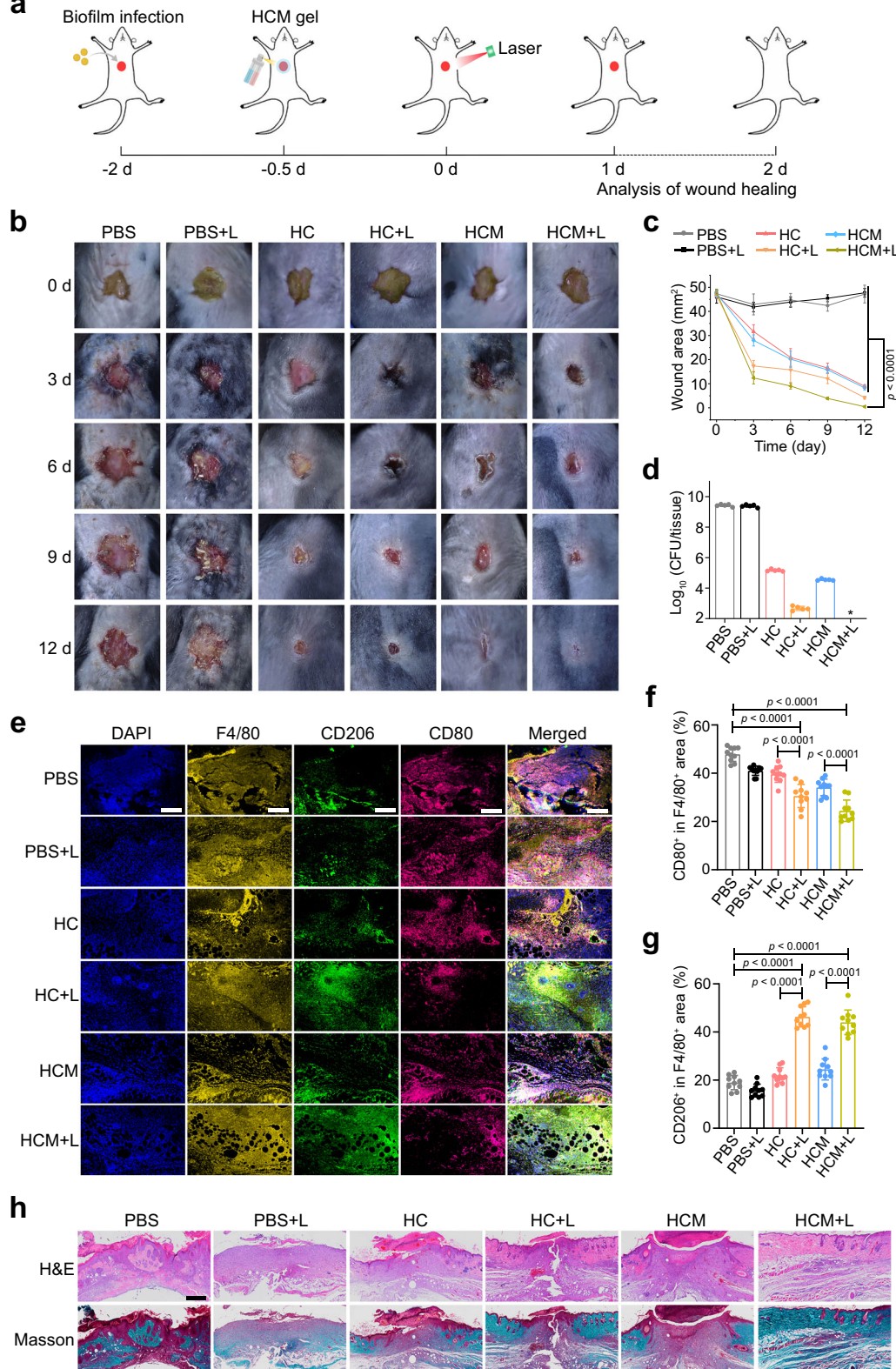

**Fig. 8 Treatment of MRSA biofilm infected wounds in diabetic mice by HCM gel. a** Schematic illustration of the experimental procedure. Photographs (**b**) and wound area (**c**) of the infected wounds after treatment ($n = 5$ biologically independent samples; mean ± SD). **d** Colony numbers of bacteria in infected wounds at 12 d post-treatment ($n = 5$ biologically independent samples; mean ± SD). **e** Immunofluorescence images of F4/80 (yellow), CD206 (green), and CD80 (violet) in infected wounds at 4th d post-treatment. Scale bar is 200 μm. Percentage of CD80$^+$ (**f**) and CD206$^+$ (**g**) area in F4/80$^+$ area calculated from immunofluorescence images ($n = 10$ biologically independent samples; mean ± SD). **h** H&E and Masson's trichrome-stained images of infected tissues at 12 d post-treatment. Scale bar is 500 μm. Statistical significance was analyzed via one-way ANOVA with a Tukey post-hoc test. Source data are provided as a Source Data file.

chemotherapy in both subcutaneous biofilm-infected mice (5.9 log) and biofilm-infected wounds in diabetic mice (8.0 log), indicating the effectiveness of the hypoxia potentiation strategy.

In biofilm-infected tissues, macrophages play important regulatory roles in all stages of tissue repair[56]. Although the immediate inflammatory response is essential for bacterial inactivation, the existence of bacterial biofilms usually causes persistent inflammation, which seriously retards tissue healing[10,12,13]. The polarization of macrophages into anti-inflammatory (M2) phenotype is desired for the healing of biofilm-infected tissues[13,46,56]. In this work, M2-like macrophages significantly increased in infected tissues after PDT-activated chemotherapy, which can greatly facilitate inflammation inhibition and tissue healing.

Effective treatment of biofilm-related infections remains a great challenge owing to the drug-tolerant bacteria in biofilms. Although PDT-activated chemotherapy has shown promising results both in vitro and in vivo, the poor penetration depth of the visible light limits its further application for deep tissue infections[3]. Thus, PDT-activated chemotherapy may be used for superficial tissue infections, such as diabetic foot. The combination of MNZ with other oxygen-dependent therapies with deep tissue penetration, such as sonodynamic therapy, may overcome this issue.

The PDT-activated chemotherapy can efficiently treat bacterial biofilm infections and promote tissue healing by facilitating macrophage polarization toward the M2-like phenotype. Hence, this work establishes a promising strategy to deal biofilm-related infections based on hypoxia potentiation, and sheds light on the rational design of therapeutic nanoagents for effective management of the bacterial biofilm-related infections.

## Methods

**Materials**. Metronidazole (Pharmaceutical Secondary Standard), vancomycin (Van, >90%), 1-(3-dimethylaminopropyl)−3-ethylcarbodiimide hydrochloride (EDC, >99%), N-hydroxysulfosuccinimide sodium salt (NHS, >98%), hyaluronidase (Hyal, 400 units/mg), fibrinogen, thrombin (~120 units/mg), dimethyl sulfoxide (DMSO, >99.9%), dimethylformamide (DMF, >99.9%), 9, 10-anthracenediyl-bis(methylene)-dimalonic acid (ABDA, >90%), and ethylenediamine (>99.5%) were obtained from Sigma-Aldrich; sodium hyaluronate (200 kDa) was purchased from SunlidaBio; and Ce6 (>99%) was bought from JenKem Technology.

**Preparation of HA-Ce6-MNZ nanoparticles (HCM NPs)**. Firstly, the sodium hyaluronate was dissolved in PBS (10 mM, pH 7.4) to form homogeneous HA solution (50 mg/mL). EDC (16.7 mg) and NHS (12.1 mg) were dissolved in PBS (10 mM, pH 7.4, 6 mL), which was mixed with 4 mL HA solution and stirred for 30 min. Then, ethylenediamine (0.2 g, dispersed in PBS) was mixed with the above solution for 12 h at room temperature (RT), dialyzed against PBS for 2 d, and further freeze-dried to obtain A-HA. EDC (1.6 mg), NHS (1.3 mg), and Ce6 (20 mg) were dissolved in DMSO solution (10 mL) and stirred for 30 min, followed by adding 10 mL A-HA solution (DMF: $H_2O$ = 1:1, 4 mg/mL) and reacting for 12 h at RT. The reaction mixture was diluted with 40 mL PBS, ultrasonicated for 15 min, and dialyzed against PBS for 2 d to form HC NPs. Subsequently, 10 mg freeze-dried HC NPs were dissolved in 10 mL DMSO-$H_2O$ solution (1:1) and mixed with MNZ (100 mg) for 12 h at RT, then dialyzed against PBS for 2 d. In this process, the DMSO was gradually removed from the solution and the solubility of Ce6 in HC NPs and MNZ decreased in the meantime, which induced their aggregation and the formation of HCM NPs. The hydrodynamic size and zeta potential of HCM NPs were measured on a ZetaPALS Potential Analyzer (Brookhaven Instruments, USA) in PBS at RT. The amount of Ce6 and MNZ were determined by UV-vis-NIR spectroscopy on a UV-3600 spectrophotometer (Shimadzu, Japan). The grafting ratio of Ce6 in HA-Ce6 is about 6.93%. The encapsulation efficiency of Ce6 and MNZ is about 24.64% and 0.35%, respectively.

**Preparation of HCM gel**. To form the HCM gel, PBS buffers containing HCM NPs (Ce6: 200 μg/mL, MNZ: 100 μg/mL) and fibrinogen (10 mg/mL) were mixed as solution A. Thrombin solution (10 NIH U/mL) in CaCl₂ (0.1 M) was used as solution B. After spraying solution A and solution B at an equal volume by using a dual-cartridge sprayer, the HCM gel was formed in several minutes.

**Hyal-responsive drug release of HCM NPs**. HCM NPs (Ce6: 40 μg/mL; MNZ: 20 μg/mL) were dispersed in PBS with different pH values (7.4 and 5.5) with or without Hyal (200 unit/mL) and gently shaken at 37 °C. The Ce6 and MNZ

released from HCM NPs were separated from the mixtures by ultrafiltration. The released Ce6 and MNZ were quantified by using UV-vis-NIR absorption spectroscopy.

**Detection of singlet oxygen**. The ABDA was used to detect $^1O_2$. Briefly, different agents and ABDA (10 μg/mL) were dissolved in PBS and exposed with 635 nm laser (20 mW/cm²) for different times. The $^1O_2$ can react with ABDA, which can be monitored via the decrease in absorbance at 380 nm.

**Cell culture**. Human normal liver (L-O2) cells and mouse smooth muscle cells (SMCs) were purchased from KeyGen BioTech. These cells were cultured in Dulbecco's modified Eagle's medium (DMEM) containing 10% fetal bovine serum (FBS, Gibco) at 37 °C under 5% $CO_2$.

**Bacterial culture**. Methicillin-resistant *Staphylococcus aureus* (MRSA, ATCC43300) were grown in chemically defined medium[57], and then incubated in Luria-Bertani (LB) medium (containing 1% glucose, $10^7$ CFU/mL) at 37 °C for 2 d to form MRSA biofilms.

**Cytotoxicity of HCM NPs**. L-O2 cells were grown in 96-well plates with $10^4$ cells per well for 1 d. Then, L-O2 cells were incubated with HCM NPs dispersions with different concentrations for 1 d. Cell viability was determined by using LDH Cytotoxicity Colorimetric Assay Kit (BioVision).

**Evaluation of the nitroreductase activity of MRSA**. The activity of nitroreductase from MRSA in biofilms was analyzed according to the previous work[36]. Briefly, MRSA biofilms were incubated under hypoxic (5% $O_2$) or normoxic conditions for 48 h. The MRSA were collected, washed three times with Tris-HCl buffer (20 mM, pH 7.6) at 4 °C, incubated with lysostaphin (50 μg/mL) and deoxyribonuclease I (DNase I, 50 μg/mL) for 30 min at 37 °C, respectively, and centrifuged (13000 × g) for 0.5 h at 4 °C. The nitroreductase activity in cell extracts was determined by using nitrofurantoin (50 μM) and nitrofurazone (50 μM) as substrates, and nicotinamide adenine dinucleotide phosphate (NADPH, 250 μM) or nicotinamide adenine dinucleotide (NADH, 250 μM) as electron donors. The absorbance at 400 nm for nitrofurazone and absorbance at 420 nm for nitrofurantoin were measured by using a microplate reader (PowerWave XS2, BioTek) to evaluate the nitroreductase activity.

**In vitro anti-biofilm effect**. Preformed MRSA biofilms were incubated with the saline solutions of MNZ (25 μg/mL), Van (50 μg/mL), HC (Ce6: 50 μg/mL), and HCM (Ce6: 50 μg/mL, MNZ: 25 μg/mL) for 6 h, respectively. The Calcein-AM was used to stain the viable bacteria inside MRSA biofilms after various treatments. To observe the structure of biofilm, MRSA biofilms were fixed with formalin for 10 min and washed with saline three times. After natural drying, the fixed MRSA biofilms were stained with crystal violet solution (0.02%) for 0.5 h, and imaged by a fluorescence microscope (Olympus IX71). To calculate the colony numbers of MRSA in biofilms, the samples were sonicated in saline for 5 min and tested by plate counting method. To study the long-term inhibition by HCM NPs, MRSA biofilms were treated by Van (50 μg/mL), HC (Ce6: 50 μg/mL), and HCM NPs (Ce6: 50 μg/mL, MNZ: 25 μg/mL) dispersed in chemically defined medium with or without laser irradiation, and further incubated for 72 h to evaluate the anti-biofilm efficacy by using the crystal violet staining method.

**SEM imaging of MRSA biofilms**. Following various treatments, MRSA biofilms were fixed by 2.5% glutaraldehyde solution for 30 min. Then, the samples were dehydrated by gradient ethanol solutions (15%, 30%, 50%, 75%, and 100%) for 15 min, respectively. The samples were sputter-coated with gold and imaged by using a scanning electron microscope (Hitachi S4800).

**Assessment of the hypoxia level of MRSA biofilms**. MRSA biofilms were grown on Petri dishes and incubated with HCM NPs in chemically defined medium (Ce6: 50 μg/mL, MNZ: 25 μg/mL) for 6 h. After 635 nm laser irradiation (20 mW/cm², 30 min), MRSA biofilms were stained with [Ru(dpp)₃]Cl₂ (10 μg/mL) for 6 h. Fluorescence images of MRSA biofilms were obtained using a confocal laser scanning microscope (Olympus IX81).

**In vivo fluorescence imaging of biofilm infections**. Female Balb/c mice (20 g, 6–8 weeks old) were purchased from Nanjing Junke Biological Engineering Co., Ltd. All animal procedures were performed in accordance with the Guidelines for the Care and Use of Laboratory Animals of Nanjing Tech University and approved by the Animal Ethics Committee of Nanjing Tech University. All mice were raised in 25 ± 3 °C (temperature), 60–70% (humidity), and 12 h light/dark cycle conditions. LB medium dispersions of MRSA ($10^9$ CFU/mL, with 1% glucose, 50 μL) were subcutaneously injected into the right thigh of mice and infected for 48 h to form subcutaneous biofilm infections. Fluorescence images of mice were captured on an IVIS Lumina K Series III system (Perkin Elmer) post i.v. injection of HCM NPs (Ce6: 4 mg/kg; MNZ: 2 mg/kg) at given time points.

**Treatment of subcutaneous biofilm infected mice by HCM NPs**. MRSA biofilm infected mice were *i.v.* injected with HCM NPs (Ce6 = 4 mg/kg; MNZ = 2 mg/kg) and irradiated with 635 nm laser (20 mW/cm$^2$, 30 min) at 8th h post-injection. The area of infected tissue was calculated as follows: area = (width/2 × length/2) × $\pi$. The infected tissues from mice were captured following an 8 d treatment period, and the bacteria were separated by ultrasonication for 15 min. The number of bacteria in the infected tissues was measured by the plate counting method. For histological analysis, all samples were fixed with 10% formalin, embedded in paraffin, and sliced for H&E and Masson's trichrome staining.

**Treatment of biofilm infected wounds in diabetic mice by HCM gel**. Male C57BL/6 mice (20 g, 6–8 weeks old) were intraperitoneally (*i.p.*) injected with streptozotocin (STZ) (60 mg/kg) every 3 days for 2 weeks. Blood glucose levels >16.7 mM indicated the establishment of diabetic mice[58]. A circular cut (8 mm in diameter) was excised from the back of diabetic mice, incubated with MRSA biofilm dispersions (10$^7$ CFU), and dressed with semiocclusive transparent films (Tegaderm, 3 M) for 24 h to form MRSA biofilm infected wounds. Then, the mixtures of different materials with fibrinogen and thrombin were sprayed on the wounds (Ce6 = 1 mg/kg; MNZ = 0.5 mg/kg) to form HCM gel. The wounds were irradiated with 635 nm laser (20 mW/cm$^2$, 30 min) at 12 h post-incubation. The infected tissues were harvested from mice at 12 d post-treatment and processed for histological analysis.

**Immunofluorescence imaging**. After various treatments for 4 d, the biofilm-infected tissues were harvested from mice, fixed in 10% formalin, and then embedded in paraffin. After slicing, the tissue sections were incubated with 3% methanol for 10 min and 1% BSA for 20 min. The sections were then incubated with the primary antibodies, including anti-HIF-1α (Abcam, ab16066), anti-VEGF (Abcam, ab52917), anti-F4/80 (Abcam, ab6640), anti-CD80 (Abcam, ab254579), and anti-CD 206 (Abcam, ab64693), at room temperature for 2 h. Following incubation, the sections were incubated with fluorescently labeled secondary antibodies, including Alexa Fluor594-IgG (Abcam, ab150160), FITC-IgG (Jackson ImmunoResearch, 111-095-003), and tetramethylrhodamine (TRITC)-IgG (Jackson ImmunoResearch, 115-025-062) for 1 h. All antibodies were diluted 200 times before used. After further staining with 4′, 6-diamidino-2-phenylindole (DAPI), all slices were imaged by using a digital pathological section scanner (Olympus VS200).

**Cytokine detection**. The infected tissues and serums of mice were harvested at 4th d post-treatment. The cytokine level in tissues and serums was measured by ELISA kits (IL-6 (Abcam, ab222503), IL-4 (Abcam, ab100710), Arg-1 (Abcam, ab269541), TGF-β (Abcam, ab119557), TNF-α (Abcam, ab208348), and IL-12p70 (Abcam, ab119531)) according to the manufacturer's instructions.

**Western blotting**. The proteins were isolated from biofilm-infected tissues at 4th d post-treatment. The content of protein was determined by using bicinchoninic acid protein assay (KeyGen BioTech). The gel electrophoresis and protein transformation were conducted by using western blotting kit (KeyGen BioTech) and the primary antibodies: anti-GADPH (Abcam, ab8245), anti-CD80 (Abcam, ab254579), and anti-CD 206 (Abcam, ab64693). All antibodies were diluted 200 times before used. The photographs were captured by G:BOX chemi-XR5 (Syngene) and relatively protein expression was quantified by using Gel-Pro Analyzer software.

**Statistical analysis**. All data are expressed as the mean ± standard deviation (SD). Inter-group and intra-group comparison analyses in each experiment were calculated by one- or two-way ANOVAs with a Tukey post-hoc test. All statistical analyses were carried out by using Graphpad Prism (version 8.4.0). Probability (*p*) values < 0.05 were considered statistically significant.

**Reporting summary**. Further information on research design is available in the Nature Research Reporting Summary linked to this article.

## Data availability
All the data supporting the findings of this study are available within the article and its Supplementary information files and from the corresponding author upon request. Source data are provided with this paper. The source data of Figs. 2d–f, 3c–e, h, j, 4d–f, 5e, f, h, i, 6b–i, 7d, e, g, 8c, d, f, g, and Supplementary Figs. 1, 2, 3, 4, 5, 6, 7, 8, 9b, 10b, 11, 13, 15, 16b, 17a, 18, 20b, 21, 22a, 23b are provided in Source Data file.

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

## Acknowledgements

This work was financially supported by the National Key Research and Development Program of China (2017YFA0205302 to L.Wang), Natural Science Foundation of Jiangsu Province (BK20191382 to L.Y., and BK20200710 to D.Y.), Postgraduate Research & Practice Innovation Program of Jiangsu Province (KYCX20_0793 to W.X.), Leading-edge Technology Programme of Jiangsu Natural Science Foundation (BK20212012 to L.Wang), Open Research Fund of Jiangsu Key Laboratory for Biosensors (L.Y.), and the Priority Academic Program Development of Jiangsu Higher Education Institutions (PAPD, YX030004 to L.Wang).

## Author contributions

W.X., L.Y., and L.Wang conceived the concept of the study and designed the experiments. W.X. performed the experiments and X.L. helped for the material characterizations. W.X., L.Wan, D.Y., and K.Y. performed the in vitro and in vivo experiments. H.D. and Y.M. helped and gave valuable suggestions for the in vivo experiments. W.X., L.Y., and L.Wang analyzed the experimental data and co-wrote the manuscript. All the authors discussed, commented, and agreed on the paper.

## Competing interests

The authors declare no competing interests.
