## [Peer Review File · Nature Communications]

Potentiating hypoxic microenvironment for antibiotic activation by photodynamic therapy to combat bacterial biofilm infectionsREVIEWER COMMENTS

Reviewer #1 (Remarks to the Author):

In the manuscript, photodynamic therapy (PDT)-activated chemotherapy by potentiating the hypoxia of biofilm infection microenvironment is proposed to tackle methicillin-resistant *Staphylococcus aureus* biofilm infection. PDT was used not only to eradicate MRSA biofilms in normoxic conditions, but also to potentiate the hypoxic microenvironment, which induces the anaerobic metabolism of MRSA and activates metronidazole to kill bacteria. Moreover, PDT-activated chemotherapy could polarize the macrophages to a M2-like phenotype and promote the repair of the biofilm infected wounds in mice. This biofilm infection microenvironment modulation strategy, whereby the hypoxic microenvironment is potentiated to synergize PDT with chemotherapy, provides an alternative pathway for efficient treatment of biofilm-associated infections.

There are some issues to be addressed.

1. In the preparation of A-HA, due to the two amine groups in ethylenediamine, one amine should be protected, or the HA would crosslink and form gel.
2. In line 427-429, because HC NPs are formed by self-assembly, if the NPs are dissolved in DMSO-H₂O solution, they cannot reform as illustrated in Figure 2. Please explain the formation of HA-Ce6-MNZ NPs?
3. In Fig. 3, the biofilm should be characterized by CLSM to calculate the thickness.
4. In Figure 1, the grafting ratio of Ce6 in HA-Ce6 should be calculated.
5. The language needs to be improved to be readable.

Reviewer #2 (Remarks to the Author):

The study presented by Xiu et al, demonstrates a very promising means of treating biofilms in superficial locations. Its results are very promising but there are some points that need addressing prior to publication.

- On Line 57 of the introduction, the authors state that biofilms are known to be hypoxic yet on line 75 the authors state that their therapy will induce hypoxia? Can the authors what level of oxygen depletion is necessary for a change in metabolism to be introduced? Is the metabolism changed or just the MNZ is more accessible to bacteria deeper in the biofilm post-PDT?
- Can the authors please introduce MNZ and Chlorin e6 fully. Nature comms is a broad scope journal.
- Line 82-83, "in situ sprayed fibrin gel containing HCM NPs was demonstrated to be effective in treating MRSA biofilm infected chronic wounds in diabetic mice". Can the authors please clarify if this was in combination with PDT?
- Spectral data from Figure 2 is too small to be read. I do see the supplementary data but I believe that inclusion of the drug in the same stack as the final particle will allow for a more meaningful comparison to be made.
- Can the authors please give the encapsulation efficiency and of the active ingredients please.
- poly-dispersity index should be included for particles
- Hyaluronic acid is quite rapidly turned over in mammalian tissue also. Given the biodistribution profile, can the authors please comment on any risks?
- Can the authors please clarify further how they determined that the s.c. injected MRSA was a mature biofilm? Are there any specific proteins that can be stained?
- Student's T test is not appropriate for the multi-sample analysis conducted in Figures 4 onwards and supplementary. Please re-analyse using 1- and 2-way ANOVAs and post-hoc tests relative to each figure. It is not possible gauge the impact of these studies without this (particularly so the cytokine analysis).
- Can the authors state if the spray delivery of the nanoparticles changes any parameters vs unsprayed? E.g. particles size/charge or drug release?
- The discussion only mentions one reference for other studies attempting PDT to eradicate MRSA. There appears to be quite an amount of work in the field and the authors should better highlight the

novelty of their own work.

- No ethical approval has been cited for the in vivo work. Please include this.

Minor issues.

- Figure 1 appears long before it is referenced in the text. Please reformat. Additionally, abbreviations in the figure need to be fully described in the figure legend.
- Same for Figure 2.
- Line 69, explain MNZ first time it appears in text.
- Similarly, HA-Ce6-MNZ nanoparticles need to be introduced before abbreviations can be used.

Reviewer #3 (Remarks to the Author):

Review on Manuscript NCOMMS-21-44314.

The article is well-written, the results are interesting and supported by enough experimental data. I recommend this work for publication after minor corrections.

- In several cases the size of the Figures is too small. I understand that the length of the communication is limited, but it might be worthwhile to move some of the figures to the Appendix for better transparency.
- line 410-417. The purity of the applied chemicals is missing.
- line 420. For preparation of particles the hyaluronate solution was stirred for only 30 min. This time is not enough to suitable swelling, we usually use 20-24 h. What was the experience relating to the time?
- the description of the dynamic light scattering equipment is missing. The authors measured Zeta potential values as well. Type, measuring conditions is missing.
- and I did not find information on the colloidal stability of the prepared NPs. please explain.

Response to the Reviewers' Comments

Reviewer #1

Comment: In the manuscript, photodynamic therapy (PDT)-activated chemotherapy by potentiating the hypoxia of biofilm infection microenvironment is proposed to tackle methicillin-resistant *Staphylococcus aureus* biofilm infection. PDT was used not only to eradicate MRSA biofilms in normoxic conditions, but also to potentiate the hypoxic microenvironment, which induces the anaerobic metabolism of MRSA and activates metronidazole to kill bacteria. Moreover, PDT-activated chemotherapy could polarize the macrophages to a M2-like phenotype and promote the repair of the biofilm infected wounds in mice. This biofilm infection microenvironment modulation strategy, whereby the hypoxic microenvironment is potentiated to synergize PDT with chemotherapy, provides an alternative pathway for efficient treatment of biofilm-associated infections.

There are some issues to be addressed.

Response: We express our sincere thanks to the reviewer for the positive comments and valuable suggestions, which will definitely help us further improve the quality of this work.

Question 1: In the preparation of A-HA, due to the two amine groups in ethylenediamine, one amine should be protected, or the HA would crosslink and form gel.

Response: Thank you very much for your question. To prepare amine-functionalized HA (A-HA), ethylenediamine was conjugated to HA by using amidation reaction according to previous report (*Theranostics*, 2014, 4, 1). To prevent the crosslink of HA with two amine groups in one ethylenediamine molecule, it is important to control the molar ratio of amine groups (-NH₂) of ethylenediamine and carboxyl groups (-COOH) of HA. When the amount of -NH₂ is far more excessive than -COOH, most -COOH will theoretically react with only one -NH₂, which prevents the crosslink of HA by ethylenediamine. As shown in Figure R1, with the increment of the molar ratio of -NH₂ and -COOH ($n_{\text{NH}_2} : n_{\text{COOH}}$) from 0:1 to 3:1, the product of HA and ethylenediamine gradually changed from solution form to gel form. When the molar ratio reached 6:1, the reaction mixture turned to solution form again, suggesting much excessive

ethylenediamine in the reaction can prevent the crosslink of HA to form gel. In our study, the molar ratio of $-NH_2$ and $-COOH$ is about 12.6, and the prepared A-HA was solution form without the formation of gel.

Figure R1 Photographs of HA solutions after reacted with ethylenediamine with the presence of NHS and EDC at different molar ratios of $-NH_2$ from ethylenediamine and $-COOH$ from HA.

Question 2: In line 427-429, because HC NPs are formed by self-assembly, if the NPs are dissolved in DMSO-H₂O solution, they cannot reformed as illustrated in Figure 2. Please explain the formation of HA-Ce6-MNZ NPs?

Response: Thank you very much for your question. As illustrated in Figure R2, the DMSO-H₂O solution was used to dissolve HC NPs due to the presence of both hydrophilic HA part and hydrophobic Ce6 part in HC NPs. After stirred for 12 h, the DMSO-H₂O solution of HC NPs with the presence of MNZ was dialyzed against PBS for 2 d. During this process, the DMSO was gradually removed from the solution and the solubility of Ce6 part (hydrophobic) in HC NPs and MNZ (hydrophobic) decreased in the meantime, which induced their aggregation and the formation of HA-Ce6-MNZ (HCM) NPs.

Figure R2 Schematic illustration of the preparation of HCM NPs.

To better express this point, we have modified the manuscript, as followed:

In the Methods section “**Preparation of HA-Ce6-MNZ nanoparticles (HCM NPs)**” in Page 20, we added the statement: “In this process, the DMSO was gradually removed

from the solution and the solubility of Ce6 in HC NPs and MNZ decreased in the meantime, which induced their aggregation and the formation of HCM NPs.”

Question 3: In Fig.3, the biofilm should be characterized by CLSM to calculate the thickness.

Response: Thanks a lot very much for your advice. To better indicate the thickness of the biofilm, we have used three-dimensional (3D) CLSM images of MRSA biofilms to replace the original two-dimensional (2D) CLSM images in Fig. 3f and calculated the thickness of the MRSA biofilm. As shown in Supplementary Fig. 4a, the average thickness of MRSA biofilms without any treatment was calculated to be about 17.6 μm . After incubated with HCM NPs, MRSA biofilms showed no significant change of their thickness in 24 h (Supplementary Fig. 4b).

Fig. 3f Three-dimensional (3D) confocal laser scanning microscopy (CLSM) images of MRSA biofilms after incubation with HCM NPs for different times. Scale bar is 200 μm .

Supplementary Fig. 4 Thickness of MRSA biofilms. **a**, Thickness of MRSA biofilms without any treatment (n=200). **b**, Thickness of MRSA biofilms after incubation with HCM NPs for different times (n = 20; mean ± SD).

To better express this point, we have modified the manuscript, as followed:

In Page 6, we replaced the original 2D CLSM images of MRSA biofilms in **Fig. 3f** by 3D CLSM images.

In Page S4, we supplemented the thickness of MRSA biofilms without any treatment and the thickness of MRSA biofilms after incubation with HCM NPs for different times as **Supplementary Fig. 4**.

In Page 5, we added the statement: “Besides, MRSA biofilms showed limited change of average thickness after incubated with HCM NPs for 24 h (**Supplementary Fig. 4**).”

Question 4: In Figure 1, the grafting ratio of Ce6 in HA-Ce6 should be calculated.

Response: Thank you for your suggestion. The grafting ratio of Ce6 in HA-Ce6 was calculated to be 6.93%, using the following formula:

$$\text{Grafting ratio of Ce6 (\%)} = (W_1/W_2) \times 100\%$$

where W_1 is the weight of Ce6 in HA-Ce6, and W_2 is the total weight of HA-Ce6.

To better express this point, we have modified the manuscript, as followed:

In Page 20, we added the statement: “The grafting ratio of Ce6 in HA-Ce6 is about 6.93%.”

Question 5: The language need to be improved to be readable.

Response: Thanks for your advice. We have carefully revised our manuscript following your advice. The English language of this paper was edited by LetPub, which is the flagship editorial service brand of Boston-based Accdon LLC. LetPub specializes in editorial services for the scholarly publishing community and provides research communications services, such as a scientific illustration, graphical and video abstracts, and plain language summaries (<https://www.letpub.com/>).

Reviewer #2

The study presented by Xiu et al, demonstrates a very promising means of treating biofilms in superficial locations. Its results are very promising but there are some points that need addressing prior to publication.

Response: Thanks a lot for your positive comments and helpful suggestions to further improve the quality of this manuscript. We believe that the quality of this paper could be significantly improved.

Question 1: On Line 57 of the introduction, the authors state that biofilms are known to be hypoxic yet on line 75 the authors state that their therapy will induce hypoxia? Can the authors what level of oxygen depletion is necessary for a change in metabolism to be introduced? Is the metabolism changed or just the MNZ is more accessible to bacteria deeper in the biofilm post-PDT?

Response: Thank you very much for your professional questions.

(1) For the first question “the authors state that biofilms are known to be hypoxic yet on line 75 the authors state that their therapy will induce hypoxia?”:

We are sorry for the ambiguous description of the hypoxic microenvironment of bacterial biofilms. As previously reported (Figure R3), the O₂ level in bacterial biofilms gradually decreases from the out layer to the inner layer owing to the restricted O₂ diffusion in biofilms (*AMPIS, 2017, 125, 76; Nat. Rev. Microbiol., 2016, 14, 563; Nanomedicine, 2016, 11, 873*). In the out layer of biofilms, bacteria with sufficient oxygen supply can take aerobic respiration; while in the inner layer of biofilms, bacteria with limited oxygen supply tend to adopt anaerobic metabolism to adapt the hypoxic microenvironment. Therefore, the heterogeneity of oxygen distribution in biofilms results in the heterogeneous metabolic states of bacteria in biofilms. For commonly used antibacterial drugs, their efficacy often depends on the metabolic states of bacteria, which makes a great challenge to completely eliminate bacteria with different metabolic states in biofilms. Therefore, we try to remodel the metabolic condition of facultative MRSA to activate the MNZ, which is only effective for anaerobic bacteria, by potentiating the hypoxic microenvironment. In this work, we take PDT not only as an antibacterial mode but also as a method to deplete the O₂ in MRSA biofilms to further

potentiate the hypoxic microenvironment, which may induce the change of metabolic state of bacteria within biofilms and enhance the antibacterial efficiency of MNZ.

Figure R3 Model of growth and activity of bacteria in biofilms. (A) Cross-section of structured biofilms consisting of bacterial cells embedded in an extracellular polymer matrix (EPS). (B) The oxygen conditions in an *in vitro* biofilm, going from high concentration of O₂ in the bulk medium surrounding the biofilm and depletion with depth in the biofilm. (C) Spatial heterogeneity in growth rate in biofilms. Bacteria close to the surface of the biofilm grow fast, while bacterial growth becomes increasingly limited with depth in the biofilm. (adapted from *AMPIS*, 2017, 125, 76).

(2) For the second question “what level of oxygen depletion is necessary for a change in metabolism?”:

As previously reported, the nitroreductase activity in bacteria closely relates to the O₂ level in the surrounding environment (*Mol. Microbiol.*, 1998, 28, 383; *FEBS J.*, 2009, 276, 3354). To investigate the influence of O₂ level on the metabolism of MRSA, the nitroreductase activity of MRSA under different O₂ levels was evaluated. Since nitrofurantoin can be reduced by nitroreductase with the present of NADPH, it was used as an indicator to measure the change of nitroreductase activity in MRSA (*J. Bacteriol.*, 2009, 191, 3403). After incubation with the extracts of the MRSA grown at different O₂ levels, the nitrofurantoin showed gradually increased reduction with the decrement of O₂ level (Figure R4), indicating the nitroreductase activity of MRSA significantly increased with the decrement of O₂ level.

As a prodrug, MNZ can be reduced by nitroreductase to amine derivatives, which can inactivate the bacteria by damaging their DNA helix (*Antimicrob. Agents Chemother.*, 1976, 10, 476; *J. Antimicrob. Chemother.*, 2018, 73, 265). In this work, the bacterial inactivation efficiency of MNZ under different O₂ levels was studied. As

shown in Supplementary Fig. 9, the inactivation efficiency of MNZ was gradually increased with the decrement of O₂ level, and about 55% MRSA were inactivated by MNZ (50 µg/mL) when the O₂ was 5%. These results indicate that increased nitroreductase activity of MRSA in hypoxic condition can increase the MRSA inactivation efficiency of MNZ with the decrement of O₂ level.

Figure R4 Absorbance of nitrofurantoin at 420 nm (OD420) after incubated with cell extracts of MRSA cultured under different O₂ levels (n = 3; mean ± SD).

Supplementary Fig. 9 Bacterial inactivation efficiency of MNZ under different O₂ levels. **a**, Photographs of the MRSA colony grown on LB plates after incubated without or with MNZ (50 µg/mL) for 24 h under different O₂ levels. **b**, Bacterial viability of MRSA after incubated without or with MNZ (50 µg/mL) under different O₂ levels (n = 3; mean ± SD). Statistical significance was analyzed via two-way ANOVA with a Tukey post-hoc test. **p*<0.05, ***p*<0.01.

(3) For the third question “Is the metabolism changed or just the MZN is more accessible to bacteria deeper in the biofilm post-PDT?”:

Firstly, the Transwell insert was used to investigate whether the penetration of MNZ from HCM NPs within MRSA biofilms can be influenced by PDT. The experimental conditions were the same as the therapeutic process described in ‘*In vitro anti-biofilm effect*’ section in Page 21. As shown in Supplementary Fig. 10, about 74.3% of MNZ was collected in receiver plate after HCM NPs incubated with MRSA biofilms without laser irradiation (without PDT), while about 74.5% of MNZ was collected for the group with laser irradiation (with PDT), suggesting the PDT process have limited influence on the penetration of MNZ within MRSA biofilms.

Secondly, the oxygen depletion by PDT for potentiating hypoxic microenvironment of MRSA biofilms was also studied. [Ru(dpp)₃]Cl₂ was used as a luminescent oxygen sensor to detect the hypoxic level within biofilms according to the reported method (*Nat. Nanotechnol.*, 2017, 12, 378). As shown in Fig. 4b, the fluorescence intensity of MRSA biofilms stained by [Ru(dpp)₃]Cl₂ significantly increased after PDT, suggesting the PDT process can greatly potentiate the hypoxic level within MRSA biofilms. As mentioned above, the potentiated hypoxic level can improve the nitroreductase activity of MRSA and subsequently enhance the bacterial inactivation efficiency of MNZ. To further confirm this in biofilms, MNZ was used to treat MRSA biofilms under different O₂ levels. As illustrated in Supplementary Fig. 11, MNZ displayed much higher bacterial inactivation efficiency under hypoxic conditions than the normoxic condition, indicating the potentiated hypoxic microenvironment can significantly improve the antibacterial activity of MNZ in biofilms.

The above results indicate that PDT-potentiated hypoxia in MRSA biofilms can change the metabolism of MRSA and further activate MNZ for better inactivation of MRSA in biofilms. In addition, the penetration of MNZ was not significantly enhanced by the PDT process, which showed limited contribution to the improved inactivation efficiency of MRSA by MNZ in biofilms.

Supplementary Fig. 10 Penetration of MNZ within MRSA biofilms. **a**, Schematic illustration for the penetration of MNZ in MRSA biofilms grown on Transwell inserts after incubation with HCM NPs. **b**, Percentage of MNZ from HCM NPs penetrated within MRSA biofilms with or without laser irradiation (635 nm, 20 mW/cm², 30 min) for different times (n = 3; mean ± SD).

Fig. 4b Three-dimensional (3D) CLSM images of [Ru(dpp)₃]Cl₂ stained MRSA biofilms treated by HCM NPs with or without laser irradiation (635 nm, 20 mW/cm², 30 min). Scale bar is 200 μm.

Supplementary Fig. 11 MRSA viability in biofilms after incubation with MNZ at different concentrations under different O₂ levels (n = 3; mean ± SD). Statistical significance was analyzed via two-way ANOVA with a Tukey post-hoc test. *****p*<0.0001.

To better express this point, we have modified the manuscript, as followed:

In Page 2, we added the statement: “In bacterial biofilms, the O₂ level gradually decreases from the outer layer to the inner layer owing to the limitation of O₂ diffusion in biofilms.”

In Page 2, we added the statement: “In the outer layer of biofilm, the bacteria exhibit metabolically active state owing to sufficient oxygen, while the bacteria located in the inner layer of biofilm with limited oxygen supply usually display metabolically less active state.”

In Page S7, we added the study of bacterial inactivation efficiency of MNZ for MRSA under different O₂ levels as **Supplementary Fig. 9**.

In Page S7, we added the characterization of MNZ penetration within MRSA biofilms with or without PDT as **Supplementary Fig. 10**.

In Page S8, we added the MRSA inactivation efficiency in biofilm by MNZ under different O₂ levels as **Supplementary Fig. 11**.

In Page 9, we added the statement: “To investigate the influence of PDT on the MNZ penetration, the permeability of MNZ in MRSA biofilms with or without PDT was studied by using Transwell inserts (**Supplementary Fig. 10a**). As shown in **Supplementary Fig. 10b**, about 74.3% of MNZ was collected in receiver plate after HCM NPs incubated with MRSA biofilms without laser irradiation, while about 74.5% of MNZ was collected for the group with laser irradiation, suggesting the PDT process had limited influence on the penetration of MNZ within MRSA biofilms. Besides, MNZ displayed much higher bacterial inactivation efficiency under hypoxic conditions than the normoxic condition (**Supplementary Fig. 11**), indicating the potentiated hypoxic microenvironment can significantly improve the antibacterial activity of MNZ in biofilms. The above results indicate that PDT-potentiated hypoxia in MRSA biofilms can change the metabolism of MRSA and further activate MNZ for better inactivation of MRSA in biofilms. In addition, the penetration of MNZ was not significantly enhanced by the PDT process, which showed limited contribution to the improved inactivation efficiency of MRSA by MNZ in biofilms.”

Question 2: Can the authors please introduce MNZ and Chlorin e6 fully. Nature comms is a broad scope journal.

Response: Thank you very much for your advice. We are sorry for the negligence.

MNZ is an antimicrobial agent that has been widely used to treat anaerobic bacterial infections in clinic (*J. Antimicrob. Chemother.*, 2018, 73, 265). As a prodrug, MNZ could be reductively activated under low oxygen level by forming imidazole fragments with high cytotoxicity toward anaerobic bacteria. However, MNZ has been reported to have neglectable antibacterial activity for the facultative anaerobes (*Clin. Infect. Dis.*, 2010, 50, 16).

As a second-generation photosensitizer, Ce6 can transfer oxygen (O₂) into singlet oxygen (¹O₂) under laser irradiation, which can efficiently damage the vital biomolecules of tumor cells and bacteria. Ce6 has been approved by U.S. Food and Drug Administration (FDA) for the photodynamic therapy (PDT) of oesophagus, bladder, skin, head and neck cancers (*Technol. Cancer Res. Treat.*, 2005, 4, 283; *Int. J. Clin. Exp. Med.*, 2014, 7, 4867). Besides, Ce6 has also been studied as a PDT agent to treat bacterial infections, which can kill wide-spectrum of bacteria without the drug-resistant issues (*Front. Microbiol.*, 2018, 9, 1299; *ACS Nano*, 2020, 14, 347; *Nano Res.*, 2022, 15, 1626).

Based on reviewer's advice, we have modified the manuscript, as followed:

In Page 18, we made a revision: "MNZ is an antimicrobial agent that has been widely used to treat anaerobic bacterial infections in clinic"

In Page 18, we added the statement: "As a second-generation photosensitizer, Ce6 can transfer O₂ into ¹O₂ under laser irradiation, which can efficiently damage the vital biomolecules of tumor cells and bacteria. Ce6 has been approved by U.S. Food and Drug Administration (FDA) for the photodynamic therapy (PDT) of oesophagus, bladder, skin, head and neck cancers. Besides, Ce6 has also been studied as a PDT agent to treat bacterial infections, which can kill wide-spectrum of bacteria without the drug-resistant issues"

Question 3: Line 82-83, "in situ sprayed fibrin gel containing HCM NPs was demonstrated to be effective in treating MRSA biofilm infected chronic wounds in diabetic mice". Can the authors please clarify if this was in combination with PDT?

Response: Thanks for your professional suggestion. We are sorry for the omission. The PDT process is involved in the treatment of MRSA biofilm infected chronic wounds in diabetic mice by using fibrin gel containing HCM NPs.

To better express this point, we have modified the manuscript, as followed:

In Page 3, we made a revision: “Moreover, *in situ* sprayed fibrin gel containing HCM NPs was demonstrated to be effective in treating MRSA biofilm infected chronic wounds in diabetic mice by PDT-activated chemotherapy”.

Question 4: Spectral data from Figure 2 is too small to be read. I do see the supplementary data but I believe that inclusion of the drug in the same stack as the final particle will allow for a more meaningful comparison to be made.

Response: Thanks for your kind suggestion. The Fig. 2d-f have been enlarged. In addition, the Fourier transformed infrared (FT-IR) spectra of Ce6 and MNZ were supplemented to Fig. 2e from Supplementary Fig. 1 for better comparison with HA, HC NPs, and HCM NPs.

Fig. 2e Fourier transformed infrared (FT-IR) spectra of Ce6, MNZ, HA, HC NPs, and HCM NPs.

Question 5: Can the authors please give the encapsulation efficiency and of the active ingredients please.

Response: Thanks for your suggestion. The encapsulation efficiency of the active ingredients (Ce6 and MNZ) was calculated according to the following equations:

$$\text{Encapsulation efficiency (\%)} = W_1/W_t \times 100\%$$

where W_1 represents the weight of loaded drugs in HCM NPs, W_t represents the total weight of drugs initially added. The encapsulation efficiency of Ce6 is calculated to be about 24.64%, and the encapsulation efficiency of MNZ is about 0.35%.

To better express this point, we have modified the manuscript, as followed:

In Page 20, we added the statement: “The encapsulation efficiency of Ce6 and MNZ is about 24.64% and 0.35%, respectively”.

Question 6: poly-dispersity index should be included for particles

Response: Thanks for your suggestion. The poly-dispersity index of HC NPs and HCM NPs is 0.22 and 0.31, respectively.

To better express this point, we have modified the manuscript, as followed:

In Page 4, we added the statement: “The polydispersity index of HC NPs and HCM NPs is approximately 0.22 and 0.31, respectively”.

Question 7: Hyaluronic acid is quite rapidly turned over in mammalian tissue also. Given the biodistribution profile, can the authors please comment on any risks?

Response: Thank you very much for your advice. Previous works indicated that HA is non-antigenic and non-immunogenic, owing to its high structural homology across species and weak interaction with blood components (*Biomaterials*, 2006, 27, 1416). The removal of HA from the circulation is very efficient, and most HA can be gradually degraded and eliminated through renal clearance, hepatic elimination and pulmonary excretion (*Vet. Med.*, 2008, 53, 397; *J. Hepatol.*, 1989, 9, 177; *Adv. Drug Delivery Rev.*, 1991, 7, 221). In our work, HA were conjugated with Ce6 and loaded with MNZ to form HCM NPs for treatment of MRSA biofilm infections in both subcutaneous infection mice model and wound infection mice model.

For the subcutaneous infection mice model, the *in vivo* fluorescence imaging was used to investigate the distribution of HCM NPs in mice after the intravenous (*i.v.*) injection. As shown in Supplementary Fig. 16, the obvious fluorescence signal of HCM NPs could be observed in the liver, lung, and kidney of mice at 4 h after *i.v.* injection, and the highest fluorescence signal was found in the infected tissue. The fluorescence of HCM NPs in the liver, lung, kidney, and infected tissue gradually decreased with time and almost disappeared at 48 h post-injection. These results indicate that HCM NPs can be excreted from liver, lung, and kidney by mice, and show limited retention in mice at 48 h after *i.v.* injection.

Supplementary Fig. 16 Retention and distribution of HCM NPs in MRSA biofilm infected mice after *i.v.* injection. **a,b**, Fluorescence images (**a**) and average fluorescence intensity (**b**) of mice major organs (heart (1), liver (2), spleen (3), lung (4), kidney (5)) and biofilm infected tissues (6) after *i.v.* injection of HCM NPs (Ce6 = 4 mg/kg; MNZ = 2 mg/kg) at different times (n = 3; mean ± SD).

For the wound infection mice model, the *in vivo* fluorescence imaging was also used to investigate the retention of HCM NPs in fibrin gel after the treatment of biofilm infected wounds in mice. As shown in Supplementary Fig. 23, the fluorescence signal in wounds gradually increased and peaked at 24 h after HCM gel treatment. Then, the fluorescence signal gradually decreased and almost disappeared at 144 h, indicating that HCM NPs could be gradually eliminated during wound healing process.

Supplementary Fig. 23 Fluorescence imaging of biofilm infected wounds in mice after treated with HCM gel. **a,b**, Fluorescence images (**a**) and fluorescence intensity (**b**) of biofilm infected wounds in mice after treated with HCM gel for different times (n = 3; mean ± SD).

To better express this point, we have modified the manuscript, as followed:

In Page S11, we supplemented the results of retention and distribution of HCM NPs in MRSA infected mice after *i.v.* injection as **Supplementary Fig. 16**.

In Page S15, we added the fluorescence imaging images of biofilm infected wounds in mice after treated with HCM gel as **Supplementary Fig. 23**.

In Page 16, we added the statement: “In addition, the infected wound of mice showed no observable retention of HCM NPs at 6 d post-treatment (**Supplementary Fig. 23**).”

Question 8: Can the authors please clarify further how they determined that the *s.c.* injected MRSA was a mature biofilm? Are there any specific proteins that can be stained?

Response: Thank you very much for your question. Compared with planktonic bacteria, the bacteria in biofilm can secrete extracellular polymeric substances (EPS) to protect the biofilm from environmental stress, including polysaccharides, extracellular DNA, and proteins, which can be used to differentiate the bacteria in biofilm form and planktonic form. As previously reported, fibronectin binding protein (*fnb*), elastin binding protein (*ebps*), and polysaccharide intercellular adhesion (PIA) production-related proteins play significant roles in MRSA biofilm formation (*Int. J. Mol. Sci.*, 2018, 19, 3487; *Infect., Genet. Evol.*, 2013, 18, 106). However, we can't find an ideal method for direct staining these proteins. Instead, many research groups used real-time quantitative polymerase chain reaction (qPCR) to investigate the expression level of related genes to determine the formation of *Staphylococcus aureus* biofilms (*J. Dairy Sci.*, 2014, 97, 6129; *Infect., Genet. Evol.*, 2013, 18, 106). Therefore, to verify whether the subcutaneously (*s.c.*) injected MRSA can form a mature biofilm in mice model, the expression level of biofilm formation related genes, including *fnbB*, *ebps*, intercellular adhesion biofilm required genes (*icaC* and *icaD*, associated with PIA production), were investigated by using qPCR. As shown in Supplementary Fig. 13, the expression levels of *ebps*, *fnbB*, *icaC*, and *icaD* RNA from MRSA in infected tissues showed significant increase, compared with planktonic MRSA, suggesting the successful formation of MRSA biofilms after *s.c.* injection in mice for 2 d.

Supplementary Fig. 13 Expression levels of *ebps*, *fnbB*, *icaC*, and *icaD* transcripts from MRSA in biofilm *in vitro* and infected tissues versus planktonic MRSA (n = 3; mean ± SD). Statistical significance was analyzed via two-way ANOVA with a Tukey post-hoc test. * $p < 0.05$, ** $p < 0.01$, *** $p < 0.001$, **** $p < 0.0001$.

Supplementary Table 1

Sequences of oligonucleotide primers used for qPCR according to previous literature (*Infect., Genet. Evol., 2013, 18, 106*).

Genes	Nucleotide sequence of primers (5'-3')	Accession numbers	Annealing temperature	Amplicon size (bp)
ebps	5-GGTGCAGCTGGTGCAATGGGTGT-3	U48826.2	60°C	191
	5-GCTGCGCCTCCAGCCAAACCT-3			
fnbB	5-ACGCTCAAGGCGACGGCAAAG-3	X62992.1	60°C	197
	5-ACCTTCTGCATGACCTTCTGCACCT-3			
icaC	5-CTTGGGTATTTGCACGCATT-3	AF086783	60°C	209
	5-GCAATATCATGCCGACACCT-3			
icaD	5-ACCCAACGCTAAAAATCATCG-3	AF086783	60°C	211
	5-GCGAAAATGCCCATAGTTTC-3			
16sRNA	5-GGGACCCGCACAAGCGGTGG-3	L37597.1	60°C	191
	5-GGGTTGCGCTCGTTGCGGGA-3			

Polysaccharides are also used to indicate the biofilm formation. Previous work has developed a wound blotting method for specific detection of the *S. aureus* biofilm formation in wounds by staining mucopolysaccharides (*Wound Repair Regen., 2017, 25, 131*). Briefly, a nitrocellulose membrane was used to collect surface biofilm components, and then stained mucopolysaccharides by ruthenium red to visualize the

biofilm formation. As shown in Figure R5, nitrocellulose membrane with MRSA biofilms showed obvious imprint after ruthenium red staining, while that with planktonic MRSA showed no obvious imprint. After applied with MRSA biofilm infected tissues from mice and then stained with ruthenium red, the nitrocellulose membrane showed obvious red imprint (Supplementary Fig. 14), which indicates the formation of MRSA biofilms in mice after *s.c.* injected.

Figure R5 Photograph of ruthenium red stained nitrocellulose membrane applied with planktonic MRSA (left) and MRSA biofilms (right).

Supplementary Fig. 14 Biofilm detection by using wound blotting method. **a,b**, Photographs of MRSA biofilm infected tissue (**a**) and ruthenium red stained nitrocellulose membrane (**b**) applied with infected tissue.

To better express this point, we have modified the manuscript, as followed:

In Page S10, we added the expression levels of *ebps*, *fnbB*, *icaC*, and *icaD* transcripts from MRSA in biofilm *in vitro* and infected tissues versus planktonic MRSA as **Supplementary Fig. 13**.

In Page S10, we added the detection of MRSA biofilm formation *in vivo* by using wound blotting method as **Supplementary Fig. 14**.

In Page 10, we added the statement: “The enhanced expression of biofilm-associated genes verified the formation of MRSA biofilms in infected tissues (**Supplementary Fig. 13**). The wound blotting result also indicated the formation of MRSA biofilms in this model (**Supplementary Fig. 14**)”.

Question 9: Student's T test is not appropriate for the multi-sample analysis conducted in Figures 4 onwards and supplementary. Please re-analyse using 1- and 2-way ANOVAs and post-hoc tests relative to each figure. It is not possible gauge the impact of these studies without this (particularly so the cytokine analysis).

Response: Thank you very much for your professional suggestion. We have re-analyzed the relative data by using one or two-way ANOVA with a Tukey post-hoc test.

To better express this point, we have modified the manuscript, as followed:

The Fig. 4d, Supplementary Fig. 9, Supplementary Fig. 11, and Supplementary Fig. 13 were analyzed via two-way ANOVA with a Tukey post-hoc test. We added the statement in relative figure legends: "Statistical significance was analyzed via two-way ANOVA with a Tukey post-hoc test. * $p < 0.05$, ** $p < 0.01$, *** $p < 0.001$, **** $p < 0.0001$."

The Fig. 5e, f, h, i, Fig. 6b-e, h, i, Fig. 7g, Fig. 8c, f, g, Supplementary Fig. 20 and Supplementary Fig. 21 were analyzed via one-way ANOVA with a Tukey post-hoc test. We added the statement in relative figure legends: "Statistical significance was analyzed via one-way ANOVA with a Tukey post-hoc test. * $p < 0.05$, ** $p < 0.01$, *** $p < 0.001$, **** $p < 0.0001$."

Question 10: Can the authors state if the spray delivery of the nanoparticles changes any parameters vs unsprayed? E.g. particles size/charge or drug release?

Response: Thank you for your question. To study the size and charge of HCM NPs after spray delivery, the HCM NPs were extracted from sprayed HCM gel. As shown in Supplementary Fig. 18, the hydrodynamic size and zeta potential of HCM NPs showed limited change after formation of HCM gel. The drug release rate of HCM NPs and HCM gel have also been evaluated in this work. As illustrated in Fig. 3d and e in Page 6, and Fig. 7d and e in Page 14, respectively, about 79% of Ce6 and 77% of MNZ were released from HCM NPs after incubated with Hyal under acidic condition for 12 h, while 43% of Ce6 and 45% of MNZ from HCM gel were released at the same condition, suggesting decreased drug release after HCM NPs encapsulated within HCM gel.

Supplementary Fig. 18 Hydrodynamic size and zeta potential of HCM NPs at different conditions. **a,b**, Hydrodynamic sizes (**a**) and zeta potential (**b**) of free HCM NPs and HCM NPs in sprayed gel (n = 3; mean ± SD).

Fig. 3 Release of Ce6 (**d**) and MNZ (**e**) from HCM NPs (Ce6: 40 µg/mL; MNZ: 20 µg/mL) under different conditions (n = 3; mean ± SD).

Fig. 7 Cumulative release profiles of Ce6 (**d**) and MNZ (**e**) from the HCM gel incubated under different conditions (n = 3; mean ± SD).

To better express this point, we have modified the manuscript, as followed:

In Page S12, we added the characterization of size and charge of HCM NPs within HCM gel as **Supplementary Fig. 18**.

In Page 14, we added the description: “The HCM NPs showed limited changes of the hydrodynamic size and zeta potential after forming HCM gel (**Supplementary Fig. 18**)”.

Question 11: The discussion only mentions one reference for other studies attempting PDT to eradicate MRSA. There appears to be quite an amount of work in the field and the authors should better highlight the novelty of their own work.

Response: Thanks for your kind advice. PDT is a promising therapeutic strategy for the treatment of bacterial infections. During the PDT process, photosensitizers can transfer O₂ to highly toxic reactive oxygen species (ROS), which can react with important biomolecules in cells and damage their functions. Compared with antibiotic treatment, PDT is much difficult for bacteria to develop resistance, which provides alternative solutions for drug-resistant bacteria infections (*Nat. Mater.*, 2016, 15, 529; *Adv. Funct. Mater.*, 2015, 25, 7189; *VIEW*, 2020, 2, 20200065). Although PDT has several advantages compared with antibiotic treatment, the hypoxia in bacterial biofilm infection microenvironment significantly limits its therapeutic efficiency. To improve the antibacterial efficiency of PDT, previous works mainly focused on the strategy to relieve the hypoxia during PDT, while the strategy to potentiate the hypoxia by PDT hasn't been exploited (*Adv. Funct. Mater.*, 2019, 29, 1903018; *Adv. Sci.*, 2020, 7, 2000398; *Research*, 2020, 2020, 9426453). In our work, the hypoxic microenvironment in bacterial biofilms was potentiated by PDT to induce the anaerobic metabolism of MRSA and activate metronidazole (MNZ) for bacteria killing, which provides novel PDT-activated chemotherapy for the treatment of MRSA biofilm infections.

To better express this point, we have modified the manuscript, as followed:

In Page 17, we added the statement: “PDT is a promising therapeutic strategy for the treatment of bacterial infections. During the PDT process, photosensitizers can transfer O₂ to highly toxic reactive oxygen species (ROS), which can react with important biomolecules in cells and damage their functions. Compared with antibiotic treatment, PDT is much difficult for bacteria to develop resistance, which provides alternative solutions for drug-resistant bacteria infections. Although PDT has several advantages compared with antibiotic treatment, the hypoxia in bacterial biofilm infection

microenvironment significantly limits its therapeutic efficiency. To improve the antibacterial efficiency of PDT, previous works mainly focused on the strategy to relieve the hypoxia during PDT, while the strategy to potentiate the hypoxia PDT has not been exploited. In our work, the hypoxic microenvironment in bacterial biofilms was potentiated by PDT to induce the anaerobic metabolism of MRSA and activate MNZ for bacteria killing, which provides novel PDT-activated chemotherapy for the treatment of MRSA biofilm infections.”

Question 12: No ethical approval has been cited for the in vivo work. Please include this.

Response: Thank you very much for your suggestion. The relative description was mentioned in “***In vivo* fluorescence imaging of biofilm infections**” section in Page 22: “All animal procedures were performed in accordance with the Guidelines for the Care and Use of Laboratory Animals of Nanjing Tech University and approved by the Animal Ethics Committee of Nanjing Tech University.”

Minor issues.

- Figure 1 appears long before it is referenced in the text. Please reformat. Additionally, abbreviations in the figure need to be fully described in the figure legend.
- Same for Figure 2.

Response: We are sorry for our negligence. We have reformatted these figures in the text, and fully described the related abbreviations in the related figure legends, respectively.

To better express this point, we have modified the manuscript, as followed:

In Page 3, we revised the statement: “**Fig. 1 Potentiating hypoxia by PDT for antibiotic activation to combat MRSA biofilm infections.** a Schematic illustration of the structure of HA-Ce6-MNZ nanoparticles (HCM NPs, hyaluronic acid (HA), chlorin e6 (Ce6), metronidazole (MNZ))”

In Page 5, we revised the statement: “**Fig. 2 Preparation and characterization of HCM NPs.** a Schematic illustration of the preparation of HCM NPs (amine-functionalized HA (A-HA), 1-(3-dimethylaminopropyl)-3-ethylcarbodiimide hydrochloride (EDC), N-hydroxysulfosuccinimide sodium salt (NHS))”

- Line 69, explain MNZ first time it appears in text.

Response: We are sorry for our negligence. The full description of MNZ was supplemented in Page 2.

In Page 2, we revised the statement: “Herein, we develop the hypoxia-potentiating strategy by combining the PDT and the prodrug metronidazole (MNZ) to treat bacterial biofilm infections.”

- Similarly, HA-Ce6-MNZ nanoparticles need to be introduced before abbreviations can be used.

Response: We are sorry for our negligence. The fully description of HA-Ce6-MNZ nanoparticles were supplemented before the abbreviations used in Page 2.

In Page 2, we revised the statement: “As shown in Fig. 1a, the hyaluronic acid (HA) was functionalized with chlorin e6 (Ce6) and MNZ to form HA-Ce6-MNZ nanoparticles (HCM NPs).”

Reviewer #3

Comment: The article is well-written, the results are interesting and supported by enough experimental data. I recommend this work for publication after minor corrections.

Response: Thanks a lot for your positive recommendation and invaluable suggestions to further improve the quality of this manuscript.

Question 1: In several cases the size of the Figures is too small. I understand that the length of the communication is limited, but it might be worthwhile to move some of the figures to the Appendix for better transparency.

Response: Thank you very much for your suggestion. Some of the figures were moved in supplementary information as listed:

The Fig. 2e in original manuscript was moved to Supplementary Fig. 1.

The Fig. 5d in original manuscript was moved to Supplementary Fig. 15.

The Fig. 8h-k in original manuscript was moved to Supplementary Fig. 21.

Question 2: line 410-417. The purity of the applied chemicals is missing.

Response: Thanks for your advice. We are sorry for our negligence. The purity of the applied chemicals used in this work were supplemented.

To better express this point, we have modified the manuscript, as followed:

In Page 19, we revised the “**Materials**” section as followed:

Materials. Metronidazole (Pharmaceutical Secondary Standard), vancomycin (Van, >90%), 1-(3-dimethylaminopropyl)-3-ethylcarbodiimide hydrochloride (EDC, >99%), N-hydroxysulfosuccinimide sodium salt (NHS, >98%), hyaluronidase (Hyal, 400 units/mg), fibrinogen, thrombin (~120 units/mg), dimethyl sulfoxide (DMSO, >99.9%), dimethylformamide (DMF, >99.9%), 9, 10-anthracenediyl-bis(methylene)-dimalonic acid (ABDA, >90%), and ethylenediamine (>99.5%) were obtained from Sigma-Aldrich; sodium hyaluronate (200 kDa) was purchased from SunlidaBio; and Ce6 (>99%) was bought from JenKem Technology.

Question 3: line 420. For preparation of particles the hyaluronate solution was stirred for only 30 min. This time is not enough to suitable swelling, we usually use 20-24 h. What was the experience relating to the time?

Response: Thank you very much for your question. We are sorry for that the description of the experimental details is not clear enough. In our study, the sodium hyaluronate (HA) was dissolved in PBS (50 mg/mL) as stock solution and stored at 4°C for whole night before further use. The PBS solution of HA is clear and homogeneous, suggesting the swelling time in this process is enough. As mentioned in line 420, the 30 min stirring is used to activate the carboxyl group of HA to conjugate with the amine group of ethylenediamine, rather than to swell HA.

As shown in Figure R6, the IR adsorption bands of A-HA near 1590 cm^{-1} can be assigned to the N-H bending vibration from amide bond in A-HA, suggesting the successful formation of A-HA.

Figure R6 Fourier transformed infrared (FT-IR) spectra of HA and A-HA.

To better express this point, we have modified the manuscript, as followed:

In Page 19, we made a revision: “Firstly, the sodium hyaluronate was dissolved in PBS (10 mM, pH 7.4) to form homogeneous HA solution (50 mg/mL). EDC (16.7 mg) and NHS (12.1 mg) were dissolved in PBS (10 mM, pH 7.4, 6 mL), which was mixed with 4 mL HA solution and stirred for 30 min.”

Question 4: the description of the dynamic light scattering equipment is missing. The authors measured Zeta potential values as well. Type, measuring conditions is missing.

Response: Thank you very much for your advice. We are sorry for our negligence. In this work, the dynamic light scattering (DLS) and zeta potential were measured on a

ZetaPALS Potential Analyzer (Brookhaven Instruments, USA) in PBS at room temperature.

To better express this point, we have modified the manuscript, as followed:

In Page 20, we added the statement: “The hydrodynamic size and zeta potential of HCM NPs were measured on a ZetaPALS Potential Analyzer (Brookhaven Instruments, USA) in PBS at room temperature.”

Question 5: and I did not find information on the colloidal stability of the prepared NPs. please explain.

Response: Thank you very much for your suggestion. The relative description was mentioned in Page 5: “HCM NPs showed no obvious aggregation and change of diameter after dispersed in different mediums (H₂O, phosphate buffered saline (PBS), and minimum Eagle’s medium (MEM)) for 24 h, indicating good colloidal stability (Supplementary Fig. 5).”

Supplementary Fig. 5 Colloidal stability of HCM NPs. **a,b**, Hydrodynamic sizes and photographs (inset) of HCM NPs (Ce6: 20 $\mu\text{g}/\text{mL}$; MNZ: 10 $\mu\text{g}/\text{mL}$) dispersed in different mediums (H₂O, PBS, and minimum Eagle’s medium (MEM)) for 0 h (**a**) and 24 h (**b**).

REVIEWERS' COMMENTS

Reviewer #1 (Remarks to the Author):

As the authors have addressed all the comments according to the manuscript revise, it is recommended to be accepted for the manuscript in its current state in the prestigious journal of Nature Communications.

Reviewer #2 (Remarks to the Author):

I thank the authors for robustly addressing all points raised from the previous review. I have no further points to raise and am happy for the publication to proceed.

Reviewer #3 (Remarks to the Author):

The authors made all the requested corrections.
I accept this work for publication.